# 🐍📻Mamba Modulation
# On the Length Generalization of Mamba

**Peng Lu**[*]
Université de Montréal
peng.lu@umontreal.ca

**Jerry Huang**[*]
Université de Montréal & Mila - Quebec AI Institute
jerry.huang@mila.quebec

**Qiuhao Zeng**
Western University
qzeng53@uwo.ca

**Xinyu Wang**
McGill University
xinyu.wang5@mail.mcgill.ca

**Boxing Chen**
Noah's Ark Lab
boxing.chen@huawei.com

**Philippe Langlais**
Université de Montréal
felipe@iro.umontreal.ca

**Yufei Cui**[†]
Noah's Ark Lab
yufei.cui@huawei.com

## Abstract

The quadratic complexity of the attention mechanism in Transformer models has motivated the development of alternative architectures with sub-quadratic scaling, such as state-space models. Among these, Mamba has emerged as a leading architecture, achieving state-of-the-art results across a range of language modeling tasks. However, Mambas performance significantly deteriorates when applied to contexts longer than those seen during pre-training, revealing a sharp sensitivity to context length extension. Through detailed analysis, we attribute this limitation to the out-of-distribution behavior of its state-space dynamics, particularly within the parameterization of the state transition matrix $\boldsymbol{A}$. Unlike recent works which attribute this sensitivity to the vanished accumulation of discretization time steps, $\exp(-\sum_{t=1}^{N}\Delta_t)$, we establish a connection between state convergence behavior as the input length approaches infinity and the spectrum of the transition matrix $\boldsymbol{A}$, offering a well-founded explanation of its role in length extension. Next, to overcome this challenge, we propose an approach that applies spectrum scaling to pre-trained Mamba models to enable robust long-context generalization by selectively modulating the spectrum of $\boldsymbol{A}$ matrices in each layer. We show that this can significantly improve performance in settings where simply modulating $\Delta_t$ fails, validating our insights and providing avenues for better length generalization of state-space models with structured transition matrices. Our code is available at https://github.com/gnepul-ace/mamba_modulation.

## 1 Introduction

In the new age of deep learning, the Transformers [129] architecture has spurred a new age of research into large language models (LLMs) [31, 141, 25, 92, 148] that has largely dominated the space of natural language processing (NLP) research since their introduction. Their surprising capabilities and rapid development have led to their wide application across various domains, including chatbots,

---

[*]Equal contribution.
[†]Corresponding author.

39th Conference on Neural Information Processing Systems (NeurIPS 2025).

intelligent agents, code assistants, etc. However, the Transformer comes with various deficiencies, which has led to research into alternative paradigms that seek to resolve these outstanding concerns. One of these competitors, Mamba [45, 22], is based off the state-space model (SSM) paradigm from control theory [47, 49] that have enabled the training of recurrent models that have overcome the sequential bottleneck of traditional models [109, 55, 15].

Among the various motivation for Mamba and its successors is the goal of length extrapolation, whereby a model that is initially trained on a limited context length (e.g. 2048 tokens in each training sequence) is capable of generalizing to longer sequences at test time (i.e. without further training) due to a more efficient inference-time token processing methodology. However, various works [56, 29, 57] have brought about challenges to this claim. Meanwhile, a key component in the Transformer is the position embedding, for which Rotary Position Embeddings (RoPE) [121] has been a popular choice and applied in many LLMs. Various works have studied RoPE [139, 83] and shown it to be intuitive to manipulate to extend the context window within Transformers [14, 11, 35, 96], whereas no equivalent method yet exists for Mamba-style models. A common explanation for the ability to extend this context length is through as avoiding out-of-distribution (OOD) rotation angles [83, 52] in RoPE, meaning the extended context length (OOD) can be mapped to the in-distribution (ID) context length on which models have been properly trained. However, Mamba does not utilize knowledge of token positions during training, thus making such methods broadly inapplicable.

Recent works [146, 9, 3] have meanwhile made attempts at exploring how to conduct length generalization with Mamba models. A shared feature among these is the focus on a specific input-dependent parameter, $\Delta$, which is used to discretize the underlying state-space model and control for both the state decay over time as well as the incoming input contribution to the state. Their observations rely on the implicit notion that since the duration of the time-step influences the state, it can act as a proxy to filter out (or ignore) parts of the input, or be scaled to influence the long-term information decay within the model state. However, despite the compelling intuition behind such a notion, there remains no fundamental theoretical justification for this.

In this work, we attempt to build a better understanding of how to better scale Mamba models for improved length generalization. We begin with an analysis of the model and the implicit effects this will have on the convergence behavior of the hidden state as the input length goes to infinity. From this, we identify two ways in which this process can be controlled: either through the state-transition matrix $\boldsymbol{A}$, or through the discretization time-steps $\Delta$. We then motivate why controlling for or adjusting $\boldsymbol{A}$ is more well-founded, through arguments relating the eigenvalue spectrum with long-term information decay. Through experiments on standard long-context extension settings, such as long-context language modeling and passkey retrieval, we demonstrate empirically how scaling $\boldsymbol{A}$ is more effective compared to scaling $\Delta$, in the case of both Mamba and Mamba2 models. Broadly, we summarize our contributions as follows:

I) We first provide a broader understanding of the length generalization ability of Mamba-based models via spectrum analysis of their transition matrix. We demonstrate and justify that the convergence behavior of the hidden states hinders their length generalization in Mamba models.

II) Based on our analysis, we identify how the scaling of $\boldsymbol{A}$ as opposed to the more common practice of scaling $\Delta$ is a more effective proposition.

III) Results on a series of long-context generalization tasks show such an intuition holds empirically on Mamba models, highlighting the potential benefits of using $\boldsymbol{A}$ for length generalization.

## 2 Related Works

### 2.1 Language Models and Long Contexts

Being capable of modeling long sequences is an important desiderate in various LLM applications. However, due to the quadratic complexity (relative to the sequence length) of the self-attention mechanism in Transformers, long sequence modeling requires a large computational overhead [125]. Early work in efficient Transformers [74, 147, 8, 131, 16] attempted to reduce the computational complexity of attention by inducing greater sparsity. Additional work has explored the use of linear attention [72, 86, 143, 144, 150, 87] to remove the softmax activation that induces this

quadratic complexity, however, such methods may impair performance on tasks that demand precise contextual recall [1, 133]. Furthermore, hardware optimizations for more efficient computation [22, 21, 114, 80] as well as inference-time acceleration methods [138] to reduce the computational and memory complexity of Transformers. However, a broader class of linear recurrent models [49, 45, 93, 7, 101, 102], which resemble traditional recurrent neural networks but provide an additional benefit of parallel training over the sequence elements, have emerged as an alternative for long sequences through a sub-quadratic complexity relative to sequence length as well as constant-time inference complexity.

## 2.2 Length Generalization and Extrapolation

Various restrictions on the data available for training make it difficult to directly collect data of extreme lengths (e.g., 100K+ tokens), hence there have been a great deal of efforts devoted to enabling models to generalize beyond the training length. However, various works have demonstrated the collapse of the performance [122, 83, 139], thus leaving this an open area of research. Based on the wide dominance of RoPE as the positional embedding of choice, many recent works have focused on extending the context window by scaling the rotary angles [11, 35, 96, 14] with potentially some additional tuning, enabling extension to sometimes up to $10\times$ the original training context length. Alternatively, linear recurrent models present promise through their lack of direct positional encoding; rather, a fixed-size hidden state is often utilized to maintain information from the past while the sequence is being processed. While some promise has been shown on synthetic tasks [1, 99], where these models have been shown to be able to filter out noise from the sequence while maintaining useful information within the state, these observations have not extended to tasks such as real-world long-context language tasks [56].

Yet because many existing methods relevant to Transformer length extrapolation rely explicitly on positional information, it remains an open work to find ways to enable such linear recurrent models to generalize beyond their training lengths. Alternatively, recent works [9, 146, 3] have investigated the post-hoc length extension in `Mamba` models, with a particular focus on using the discretization time-steps $\Delta_t$ for context extension. Ben-Kish et al. [9] use the value of these time-steps to 'decimate' or remove tokens from the processing of the sequence at specific layers, resulting in a shortened sequence length. Similarly, Ye et al. [146] use the value of the time-steps to filter out tokens. Azizi et al. [3] meanwhile calibrate scaling factors for these time-steps to adjust the long-term decay within the model, extending the context length. Unlike these works, we do not analyze the effect of discretization ($\Delta$) on extrapolation ability. Instead, we focus on establishing a connection between the spectral characteristics of the state transition matrix and the asymptotic convergence behavior of the hidden state as the input length approaches infinity.

## 2.3 Spectrum Analysis of Linear Recurrent Models

Previous works have provided specific analysis of the eigenvalue spectrum of linear recurrent models as a way of understanding their state dynamics and the downstream influence this can have on performance. Gu et al. [46] initially provided an understanding of the specific parameterization of the state transition matrix in SSMs, determining the necessity of a Hurwitz matrix for effective sequence modeling. Orvieto et al. [93] further demonstrated how the eigenvalues have a specific influence on state decay as well as long-term dynamics during training. Beck et al. [7] further bound the state of the recurrence, implicitly bounding the spectrum as well. Finally, Grazzi et al. [44] also recently demonstrate the importance of negative eigenvalues for state-tracking tasks.

# 3 Background

## 3.1 State-Space Models (SSMs) and `Mamba`

The SSM-based models, i.e., structured state space sequence models (S4) [49] and `Mamba` [45] are inspired by the continuous system, which maps a 1-D function or sequence $\boldsymbol{x}(t) \in \mathbb{R}^{d_m}$ to an output $\boldsymbol{y}(t) \in \mathbb{R}^{d_m}$ through a hidden state $\boldsymbol{h}(t) \in \mathbb{R}^{d_h}$. The system uses evolution parameters $\boldsymbol{A} \in \mathbb{R}^{d_h \times d_h}$, $\boldsymbol{B} \in \mathbb{R}^{d_h \times d_m}$, and $\boldsymbol{C} \in \mathbb{R}^{d_m \times d_h}$, creating a continuous system whose dynamics are governed by

$$\boldsymbol{h}'(t) = \boldsymbol{A}\boldsymbol{h}(t) + \boldsymbol{B}x(t), \quad \boldsymbol{y}(t) = \boldsymbol{C}\boldsymbol{h}(t) \tag{1}$$

The `Mamba` model uses the selective SSM blocks, which leverage the input-dependent discretization into the recurrence computation. This is done by including an input-dependent timescale parameter $\Delta(\boldsymbol{x}_t)$ to transform the continuous parameters $\boldsymbol{A}$, $\boldsymbol{B}$ to discrete parameters $\overline{\boldsymbol{A}}_t$ and $\overline{\boldsymbol{B}}_t$. We follow the official implementation of `Mamba` [45]:

$$\boldsymbol{h}_t = \overline{\boldsymbol{A}}_t \boldsymbol{h}_{t-1} + \overline{\boldsymbol{B}}_t \boldsymbol{x}_t, \qquad \boldsymbol{y}_t = \boldsymbol{C}_t \boldsymbol{h}_t, \tag{2}$$

This method uses a Zero-Order Hold (ZOH) for the matrix $\overline{\boldsymbol{A}}_t$ and a simplified Euler discretization for the matrix $\overline{\boldsymbol{B}}_t$, omiting the computation of matrix inversion for $\overline{\boldsymbol{B}}$ as required by the ZOH:

$$\overline{\boldsymbol{A}}_t = \exp\left(-\Delta_t \odot \boldsymbol{A}\right), \qquad \overline{\boldsymbol{B}}_t = \Delta_t \otimes \left((\Delta_t \boldsymbol{A})^{-1} \left(\exp\left(\Delta_t \boldsymbol{A}\right) - \boldsymbol{I}\right) \boldsymbol{B}\right), \tag{3}$$

The key improvement of `Mamba` is making the parameterization ($\overline{\boldsymbol{A}}_t$, $\overline{\boldsymbol{B}}_t$ and $\boldsymbol{C}_t$) input-dependent. Specifically, each part of them can be computed as follows:

$$\Delta_t = \mathsf{softplus}\left(\mathsf{Linear}_\Delta\left(\boldsymbol{x}_t\right)\right), \qquad \boldsymbol{B}_t = \mathsf{Linear}_B\left(\boldsymbol{x}_t\right), \qquad \boldsymbol{C}_t = \mathsf{Linear}_C\left(\boldsymbol{x}_t\right) \tag{4}$$

where $\mathsf{Linear}_\Delta$, $\mathsf{Linear}_B$ and $\mathsf{Linear}_C$ are regular linear projections, $\odot$ is the Hadamard product, $\otimes$ is the outer product, $\Delta_t \in \mathbb{R}^d_+$ and $\boldsymbol{A} = \mathsf{diag}\left(\alpha_1, \dots, \alpha_d\right)$ s.t. $\alpha_i > 0 \,\forall i \in \{1, \dots, d\}$. In `Mamba`, $\Delta$, $\boldsymbol{B}$ and $\boldsymbol{C}$ are input-dependent, such that at each time-step unique transition matrices can be used to update the system ($\boldsymbol{A}$ is left as a fixed parameter in as the dynamics of the state should be consistent across steps). This is based on the observation that some elements in a discrete sequence may not be as important as others, therefore there is an incentive to possibly update the system differently based on this factor. This results in unique update matrices at each time-step $\left(\Delta_t, \overline{\boldsymbol{A}}_t, \overline{\boldsymbol{B}}_t, \boldsymbol{C}_t\right)$, enabling the ability to solve problems that require selective processing of the sequence. In order to maintain computational efficiency $\boldsymbol{A}$ is restricted to having a diagonal structure such that only the diagonal elements of these matrices need to be stored. `Mamba2` [22] further restricts the diagonal matrix to have the form of a scalar-times-identity matrix, enabling further computational improvements.

### 3.2 Limitations of `Mamba` in Long Context

The output can be reformulated as a matrix product form as follows:

$$\boldsymbol{Y} = \begin{bmatrix} \boldsymbol{y}_1 \\ \boldsymbol{y}_2 \\ \vdots \\ \boldsymbol{y}_L \end{bmatrix} = \begin{bmatrix} \boldsymbol{C}_1 \bar{\boldsymbol{B}}_1 & 0 & \cdots & 0 \\ \boldsymbol{C}_2 \bar{\boldsymbol{A}}_2 \bar{\boldsymbol{B}}_1 & \boldsymbol{C}_2 \bar{\boldsymbol{B}}_2 & \cdots & 0 \\ \vdots & \vdots & \ddots & 0 \\ \boldsymbol{C}_L \prod_{t=2}^{L} \bar{\boldsymbol{A}}_t \bar{\boldsymbol{B}}_1 & \boldsymbol{C}_L \prod_{t=3}^{L} \bar{\boldsymbol{A}}_t \bar{\boldsymbol{B}}_2 & \cdots & \boldsymbol{C}_L \bar{\boldsymbol{B}}_L \end{bmatrix} \begin{bmatrix} \boldsymbol{x}_1 \\ \boldsymbol{x}_2 \\ \vdots \\ \boldsymbol{x}_L \end{bmatrix} = \boldsymbol{M}\boldsymbol{X} \tag{5}$$

In this formulation, each output $\boldsymbol{y}_t$ is computed as a weighted sum of all inputs, with each weight involving a product of transition matrices $\prod_{t=j+1}^{L} \bar{\boldsymbol{A}}_t$. This product term plays a crucial role in determining the influence of past states and can be further disentangled to enable fine-grained analysis, facilitating a deeper understanding of state evolution and transition behavior.

$$\prod_{t=j+1}^{L} \bar{\boldsymbol{A}}_t = \prod_{t=j+1}^{L} \exp\left(-\boldsymbol{A}\Delta_t\right) = \exp\left(-\boldsymbol{A} \sum_{t=j+1}^{L} \Delta_t\right) \tag{6}$$

Previous work [9, 146, 3] has primarily focused on analyzing the discretization step $\Delta_t$ for long context, particularly the vanishing effect of the accumulated term $\exp\left(-\sum_{t=1}^{N} \Delta_t\right)$ when $N$ is large and propose different solutions to overcome this out-of-distribution (OOD) issue. For instance, Azizi et al. [3] propose applying scalar values $s \leq 1$ across different model layers to mitigate OOD discretization steps, ensuring smaller $\Delta_t$ to prevent the vanishing issue of distant inputs. They introduce two calibration methods and demonstrate superior length generalization performance in calibrated `Mamba` models with unconstrained scaling factors. However, their work does not explain why some of resulted scaling factors $s > 1$ could still enhance generalization performance.

## 4 Spectrum-Based Analysis of `Mamba`s Length Generalization

In this section, we examine the length generalization ability of `Mamba`-based language models from the perspective of spectrum analysis of their transition matrix. Specifically, we analyze the state convergence behavior of the hidden state in `Mamba`. Based on our findings, we propose a spectrum scaling method to enhance the length generalization capability of pre-trained `Mamba` models.

## 4.1 Spectrum of `Mamba` Transition Matrix

We first visualize the spectrum of the continuous transition matrix $\boldsymbol{\Lambda} = \mathrm{diag}\left(\exp\left(-\boldsymbol{A}\right)\right)$ of `Mamba` models. The $\exp\left(-\boldsymbol{A}\right)$ parameterization guarantees all values $\lambda \in (0,1)$. We stack all 48 layers and rank eigenvalues in descending order for each row. The magnitudes (appearing to range from 0 to 1) imply that all eigenvalues $\lambda$ lie well inside the unit circle, which is critical for the stability of the dynamics governed by the transition matrix for `Mamba` training. High eigenvalue zones can be viewed as dominant temporal modes, useful for modeling long-term dependencies, especially in language or time series tasks. We also observe that low-eigenvalue regions in the transition matrix spectrum correspond to rapidly decaying modes, which specialize in modeling local dependencies and high-frequency dynamics.

The heatmap in Figure 1 reveals a consistent spectral pattern across layers: the coexistence of both large (near 1) and small (near 0) eigenvalues. Next, we establish the connection between the failure of length generalization and the spectrum of the transition matrix by showing a divergent tendency in the convergence of the state norm.

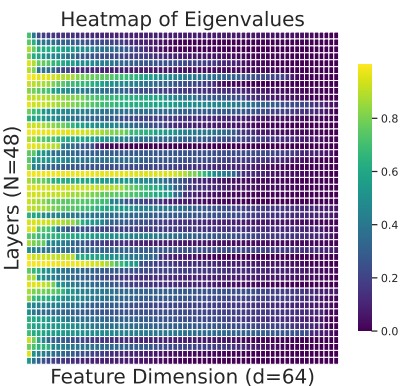

Figure 1: Heatmap of eigenvalues of $\mathrm{diag}(\exp(-\boldsymbol{A}))$ of `mamba2-1.3b`.

## 4.2 State Convergence in SSMs for Long Contexts

The previous section presented the numerical spectrum of the transition matrix $\exp(-\boldsymbol{A})$. Next, we theoretically investigate its influence on the convergence behavior of `Mamba` states. We begin by introducing the following lemma, which establishes an expected bound on the norm of inputs.

**Lemma 4.1.** *Let $\boldsymbol{B} \in \mathbb{R}^{d \times d}$ be a matrix with $\|\boldsymbol{B}\|_2 = \sigma_B$, and let $\boldsymbol{x} \in \mathbb{R}^d$ be a vector such that each entry of $\boldsymbol{x}$ satisfies $|x_i| \leq \sigma_x$. The upper bound for $\|\boldsymbol{B}\boldsymbol{x}\|_2$ is:*

$$\|\boldsymbol{B}\boldsymbol{x}\|_2 \leq \sigma_B \cdot \sigma_x \cdot \sqrt{d}. \tag{7}$$

**Theorem 4.2.** *(Convergence of State Norm with Real-Valued Diagonal Transition Matrix). Let the real-valued transition matrix $\boldsymbol{\Lambda} \in \mathbb{R}^{d \times d}$ be diagonal with eigenvalues $\lambda_i \sim \mathrm{Uniform}[\lambda_{\min}, \lambda_{\max}]$, where $0 < \lambda_{\min} < \lambda_{\max} < 1$. Consider the system dynamics: $\boldsymbol{h}_t = \boldsymbol{\Lambda}\boldsymbol{h}_{t-1} + \boldsymbol{B}\boldsymbol{x}_t$ where $\boldsymbol{x}_t$ is the input vector at time step $t$, and $\boldsymbol{B}$ is a weight matrix whose rows are independently sampled as $\boldsymbol{b} \sim \mathcal{N}\left(0, \frac{1}{2d}\boldsymbol{I}\right)$. Then, as $t \to \infty$, the expected squared norm of the state $\boldsymbol{h}_t$ converges to:*

$$\mathbb{E}[\|\boldsymbol{h}_\infty\|^2] = \frac{1}{2(\lambda_{\max} - \lambda_{\min})} \log\left(\frac{1 - \lambda_{\min}^2}{1 - \lambda_{\max}^2}\right) \cdot \mathbb{E}[\|\boldsymbol{B}\boldsymbol{x}\|^2]. \tag{8}$$

Under the setting of Theorem 4.2 (proofs in Appendix A), we consider two cases (`Mamba` and `Mamba2`) corresponding to the architectural variants of structured state-space models, each characterized by a different form of the structured transition matrix.

**Corollary 4.3** (Norm of `Mamba` State). *Suppose the diagonal entries of $\boldsymbol{\Lambda}$ are independently drawn from a uniform distribution on $[0, \lambda]$, a moderate discretized step value $\Delta$ and the system evolves as $\boldsymbol{h}_t = \boldsymbol{\Lambda}\boldsymbol{h}_{t-1} + \overline{\boldsymbol{B}}\boldsymbol{x}_t = \mathrm{diag}\left(\exp(-\Delta\alpha)\right)\boldsymbol{h}_{t-1} + \Delta\boldsymbol{B}\boldsymbol{x}_t$. Then the growth rate $\rho$ of the expected squared norm of the limiting state satisfies $\mathcal{O}\left(\frac{\Delta}{2\lambda}\log\left(\frac{1}{1-\lambda^2}\right)\right)$.*

**Corollary 4.4** (Norm of `Mamba2` State). *Suppose $\boldsymbol{\Lambda} = \lambda\boldsymbol{I} = \exp(-\Delta\alpha)\boldsymbol{I}$ is a scalar multiple of the identity matrix, where $\lambda \in (0,1)$, a moderate discretized step value $\Delta$ and the system evolves as $\boldsymbol{h}_t = \boldsymbol{\Lambda}\boldsymbol{h}_{t-1} + \overline{\boldsymbol{B}}\boldsymbol{x}_t = \exp(-\Delta\alpha) \odot \boldsymbol{h}_{t-1} + \Delta\boldsymbol{B}\boldsymbol{x}_t$. Then the convergence rate $\rho$ of the expected squared norm of the limiting state can be estimated as $\mathcal{O}(\frac{\Delta \cdot \lambda}{1-\lambda})$.*

These provide insight into the asymptotic convergence behavior of `Mamba` states as the input sequence length grows with different eigenvalues. If $\lambda \to 1$, or $\lambda \to 0$, then

$$\lim_{\lambda \to 1^-} \rho = \infty, \qquad \lim_{\lambda \to 0^+} \rho = 0 \tag{9}$$

These rates shed light on challenges in length generalization for structured state-space models (SSMs) with constrained diagonal transition matrices. In particular, both extremely large eigenvalues (approaching 1) and extremely small eigenvalues (approaching 0) can induce instability in the `Mamba` state norm as input length increasesleading to state explosion or vanishing, respectively. While tuning the discretization step $\Delta$ can help modulate the convergence rate (as suggested by Corollary 4.3 and 4.4), it does not address the root cause: the distribution of the transition matrix eigenvalues. To directly tackle this issue, we propose a *spectrum scaling* method that adjusts the spectral distribution of a pre-trained `Mamba` model by compressing large eigenvalues and inflating small ones. This rescaling aims to stabilize the state norm across longer sequences, thereby improving the models ability to generalize over input length.

### 4.3 State Norm Analysis across different input lengths

| **ProofPile** | 1K | 2K | 4K | 8K | 16K | 32K | 64K |
|---|---|---|---|---|---|---|---|
| State norm (max) | 131.4489 | 174.6176 | 892.7114 | 1330.5321 | 1141.5253 | 1130.6537 | 1157.5122 |
| State norm (min) | 0.0046 | 0.0030 | 0.0007 | 0.0007 | 0.0004 | 0.0003 | 0.0005 |
| **PG19** | 1K | 2K | 4K | 8K | 16K | 32K | 64K |
| State norm (max) | 158.7656 | 160.8465 | 166.3788 | 876.0833 | 1241.3979 | 1201.5530 | 1242.7589 |
| State norm (min) | 0.0033 | 0.0014 | 0.0050 | 0.0004 | 0.0005 | 0.0003 | 0.0002 |

Table 1: State norm statistics for ProofPile and PG19 datasets across different sequence lengths.

We conducted a series of experiments to examine how the hidden state of the SSM evolves with increasing input sequence length, as shown in Table 1. Using randomly sampled data from ProofPile and PG19, we measured the maximum difference between the largest and smallest SSM state norms across all layers of the `mamba2-1.3b` model. The observed divergence in state norm magnitude as sequence length grows provides empirical validation for the theoretical predictions outlined in the previous section.

## 5 Mamba Modulation for Length Extrapolation

In the following sections, we describe a series of experiments that we conduct to validate our previous intuitions. Appendix C provides more specific implementation details and design choices.

### 5.1 A Simple Case Analysis on Constant Scaling

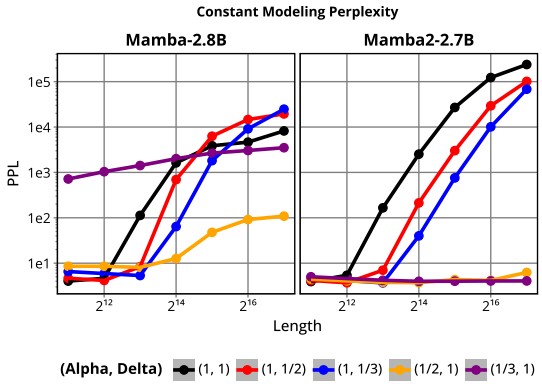

Figure 2: Language modeling perplexity on Proof-Pile after applying a constant scaling factor to either $A$ or $\Delta_t$. Lines are distinguished by their colors. $(1, 1)$ means the baseline where nothing is scaled.

To confirm our intuition, we first attempt a simple comparison between the effects of scaling $\Delta_t$ and $A$. Here, across all layers, we use a fixed, constant-valued scaling factor. We evaluate language modeling perplexity on the ProofPile dataset [40], following Peng et al. [96], across a varying number of context lengths. This method uses no tuning or training; the scaling is applied explicitly during the forward pass. Figure 2 shows these results after applying a scaling on perplexity on context lengths from 2K to 128K tokens, with scaling factors of 2 or 3 applied.

We see that scaling $A$ by a constant scaling factor is significantly better at incurring a lower perplexity, however, it remains the case that such a constant scaling factor needs to be properly tuned for, particularly in the case of `mamba-2.8b`. Given the simple setting/scenario on which we experiment, this is unsurprising; as we investigated in Section 4.2, different layers have different underlying behavior in terms of their eigenvalues, making it likely difficult

to find a constant scaling factor that can work across all layers. In the case of `mamba2-2.7b`, we can see that applying these scaling factors can significantly bound the long-context perplexity from exploding.

## 5.2 Adapting MambaExtend to Scale A

Given our observations and analysis regarding the relationship between $A$ and $\Delta_t$, a natural method against which we can compare is `MambaExtend` is a training-free method that scales the discretizations steps at each layer. For a model with $L$ layers, the objective is to learn a set of constant scaling factors for each layer $\{s_i\}_{i=1}^L$ which can be used to adjust the discretizations steps $\Delta_t$. In

---

**Algorithm 1** `MambaExtend` methodology.

1: **Input**: Model $\mathcal{M}$, calibration set $\mathcal{C}$ and function $\mathsf{CF}$
2: **Output**: Scaling factors $\boldsymbol{S} = [\boldsymbol{s}_1, \dots, \boldsymbol{s}_L] \in \mathbb{R}_+^{d_s \times L}$
3: **for** $i \leq L$ **do**
4: $\quad \boldsymbol{s}_i \leftarrow U(0, 1)$
5: **end for**
6: $\boldsymbol{S} \leftarrow \mathsf{CF}(\boldsymbol{S}, \mathcal{C}, \mathcal{M})$
7: **return** $\boldsymbol{S}$

---

general, $\boldsymbol{s}_i$ can be set to either a scalar or a vector. These scaling factors serve as learnable parameters in within the model but are consequentially tuned in a manner that does not require training any other parameters within the model. The idea of the algorithm is to take a pre-trained model along with a small set of samples for calibrating the scaling factors; depending on the setting, the calibration function can vary, with the only restriction being that the original model parameters are not modified during calibration. Appendix C.3 describes the implementation in further detail.

Although the original `MambaExtend` learns scaling factors only for $\Delta_t$, their methodology is adaptable to usage with $A$ instead; given the shared dimensionality for both $A$ and $\Delta_t$, the scaling factors can be directly used for calibrating $A$. Furthermore, this means that tuning scaling factors for $A$ does not require any additional computation, time or memory requirements as compared to tuning them directly $\Delta_t$, leading to a simple yet effective algorithm that can directly be applied to the adaptation of $A$ for long-context generalization. The following sections evaluates the performance and efficiency of tuning these scaling factors for $A$ on a number of standard settings for evaluating long-context generalization of models. As a baseline, we compare directly with `MambaExtend`.

## 6 Experiments and Results

### 6.1 Language Modeling Perplexity

We first experiment by measuring language modeling perplexity after calibrating scaling factors for either $\Delta_t$ or $A$. In this task, we use the black-box zeroth-order calibration method suggested by Azizi et al. [3]; we train a single scaling factor $s_i \in \mathbb{R}_+$ for every layer $i$ in the model. For a $L$-layer model, this means $L$ individual scaling factors are used. To calibrate, 20 samples of the corresponding context length are used. For example, for a length of 16K, 20 samples of this length are used for the calibration of the set of $s_i$. Figure 3 shows these perplexity results on a number of validation datasets, namely ProofPile [40], PG19 [103] and GovReport [59].

In particular, scaling $A$ leads to better perplexity on nearly all validation datasets, for both `Mamba` and `Mamba2` models. In many cases, this gap can be significant, particularly in the case of `mamba2-2.7b`, where the perplexity at long sequences when calibrating $\Delta_t$ explodes for all three datasets whereas calibrating $A$ can lead the model to maintain a consistent perplexity up to $1000\times$ lower.

### 6.2 Passkey Retrieval

Next, we conduct experiments on the Passkey Retrieval task, also known as the Needle-in-A-Haystack. Similar to before, we conduct this to compare the effectiveness of tuning scaling factors for $A$ as opposed to $\Delta_t$; we again conduct this experiment across different `Mamba` models. Unlike the language modeling perplexity task however, we train the model on a training set. This training set contains samples of length 4096 corresponding to the task, where the objective is standard instruction-tuning [32]. However, we freeze all parameters except the scaling parameters for each layer. For `Mamba`, it is equivalent to the number of inner state dimensions, i.e. each inner state utilizes the same scaling factor for each dimension of the SSM state. For `Mamba2`, this is the number of heads, meaning that each head shares the same scaling factor for each component of its state.

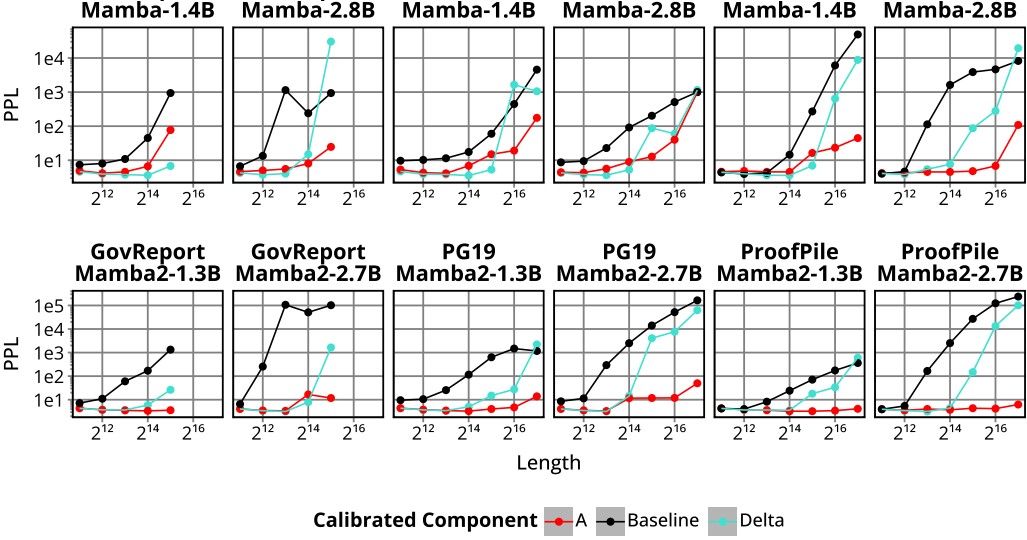

Figure 3: Model perplexity by calibrating scaling factors for either $\log(\boldsymbol{A})$ (red) or $\Delta_t$ (turquoise), across different datasets and sizes. $\boxed{\texttt{Baseline}}$ means the base model with no calibration, i.e. the model is used directly without modification.

Evaluation is conducted on a set of fixed lengths and depths to evaluate for both generalization ability as well as potential biases to relative location within the sequence. The exact setup follows from Ben-Kish et al. [9], in particular, the task comprises of a 5-digit code embedded at a random sequence depth within samples from the WikiText-103 dataset [89]. Models are deemed to have solved length/depth pair if they can correctly solve all evaluation examples, i.e. retrieve the code within the example.

Figure 4 and Appendix C.3.2 visualize these results. In particular, we see very consistent results similar to our language modeling perplexity results; for Mamba, smaller models appear to fare slightly better when trained to scale $\Delta_t$, but as the models get larger, learning to scale $\boldsymbol{A}$ closes the gap and eventually exceeds the performance of scaling $\Delta_t$. Similarly, for Mamba2 models, scaling $\boldsymbol{A}$ appears to nearly always be a more appropriate choice in comparison to scaling $\Delta_t$, as seen by a nearly constant improvement in performance on the task. Further comparing with a full-fine-tuning of the model, we observe that scaling $\boldsymbol{A}$ is as effective despite fewer parameters being trained, whereas scaling $\Delta_t$ does observe a drop-off in performance.

## 6.3 LongBench

LongBench [6] is a popular benchmark for testing the long-context abilities of LLMs, serving as a more suitable real-world benchmark on which we can explore how the scaling of $\boldsymbol{A}$ as opposed to $\Delta_t$ can influence performance. Here, we again use the zeroth-order optimization method as we used for our initial perplexity experiments. More specifically, a constant scaling factor is used for each individual layer. We compare against both the initial base model, as well as MambaExtend. Table 2 shows results on mamba2-2.7b. In particular, we show that we can increase performance by over 6% through the calibrated scaling of $\boldsymbol{A}$, with a relative improvement of nearly 10% compared to if the scaling was instead calibrated for $\Delta_t$.

Table 2: Results on LongBench [6].

| Model | Strategy | Qasper | HotpotQA | 2WMHQA | TREC | TriviaQA | LCC | RB-P | Average |
|---|---|---|---|---|---|---|---|---|---|
| | Base Model | 1.17 | 1.54 | 2.18 | 8.33 | 10.60 | 23.46 | 14.97 | 8.75 |
| mamba2-2.7b | MambaExtend | 12.53 | 1.63 | 5.99 | 24.63 | 10.33 | 23.00 | 17.09 | 13.60 |
| | Calibrated Scaling $\boldsymbol{A}$ | 12.90 | 5.69 | 11.18 | 24.32 | 10.49 | 23.36 | 16.91 | 14.98 |

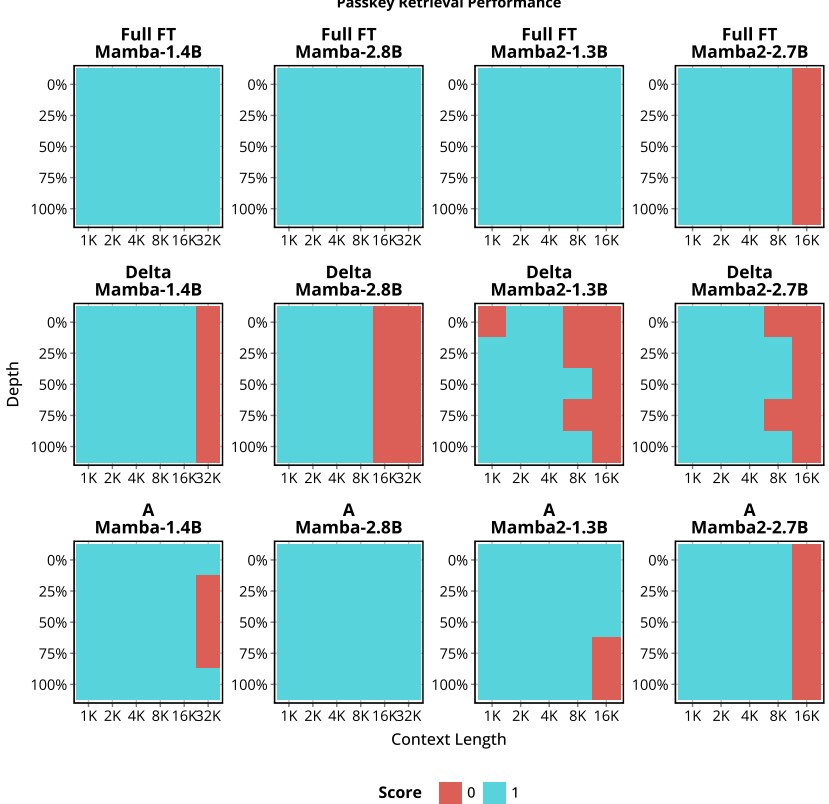

Figure 4: Passkey Retrieval performance of `Mamba` models by calibrating scaling factors for either $\log{(\boldsymbol{A})}$ or $\Delta_t$. Turquoise squares mean that the model was able to solve all examples of the given evaluation length/depth pair after tuning scaling factors, while red squares means otherwise.

Furthermore, if looking more specifically at individual tasks, there are no settings where calibrating $\Delta_t$ results in a meaningful performance increase compared to $\boldsymbol{A}$, whereas calibrating $\boldsymbol{A}$ instead appears to significantly increase performance on HotpotQA and 2WikiMultihopQA.

## 6.4    Comparison with Alternative Methods

As a final point of comparison of our proposed methodology, we compare against other proposals that have aimed towards extending the context of `Mamba`. Unlike `MambaExtend`, both of these methods use a filtering mechanism rather than directly scale $\Delta_t$; in `LongMamba` [146], channels are prevented from exponential decaying by filtering out tokens from the training sequence if the update of a specific token within the sequence $\Delta_t$ is smaller than a preset threshold. `DeciMamba` [9] instead defines *decimating layers* that directly filter out tokens that are then not passed to the following layer, significantly shortening the sequence that the last layers within the model observe. Both models require additional tuning; `LongMamba` calibrates multiple hyper-parameters to tune their filtering mechanism, while `DeciMamba` requires training the decimation layers on longer sequences.

For reasons of public code availability[3] and methodology[4], we compare `DeciMamba` against `mamba-1.4b/2.8b` and `LongMamba` against `mamba2-1.3b`. We also provide results using the initial base model, `MambaExtend`, as well as the previous two ways we tested for scaling $\boldsymbol{A}$, namely constant scaling as well as the calibrated scaling based on `MambaExtend`. Table 3 compares the effectiveness of these different methods on perplexity on the PG19 dataset. We note that in all cases, the calibrated scaling of $\boldsymbol{A}$ performs either the best or second best on all context lengths across the different tested models with marginal gaps when not the best performing method, while other

---

[3]`LongMamba` did not release their tuning code: `https://github.com/GATECH-EIC/LongMamba`
[4]`DeciMamba` only modified Mamba CUDA kernels: `https://github.com/assafbk/DeciMamba`

Table 3: Comparison of PG19 perplexity at varying lengths. Cases where scaling $A$ leads to the lowest perplexity are **bolded** and underlined when second best. If the best method does not involve scaling $A$, it is highlighted in violet.

| Model | Context Length | | | | | |
|-------|------|------|------|------|------|------|
|       | 2K | 4K | 8K | 16K | 32K | 64K |
| mamba-1.4b | | | | | | |
| Base Model | 9.67 | 10.23 | 11.43 | 17.46 | 59.77 | 444.09 |
| DeciMamba | 11.45 | 12.34 | 14.65 | 19.83 | 24.85 | 28.48 |
| MambaExtend | 4.69 | 3.89 | 3.83 | 3.55 | 5.31 | 1648.0 |
| Constant Scaling $A$ | 44.68 | 53.46 | 59.56 | 63.51 | 75.86 | 114.44 |
| Calibrated Scaling $A$ | 5.31 | 4.31 | 4.13 | 6.88 | 14.94 | **19.13** |
| mamba-2.8b | | | | | | |
| Base Model | 8.66 | 9.42 | 22.78 | 91.43 | 202.20 | 508.88 |
| DeciMamba | 11.34 | 13.45 | 15.63 | 18.34 | 21.53 | 26.54 |
| MambaExtend | 4.25 | 3.80 | 3.63 | 5.25 | 87.00 | 60.00 |
| Constant Scaling $A$ | 28.80 | 33.93 | 39.80 | 69.37 | 162.77 | 355.77 |
| Calibrated Scaling $A$ | 4.44 | 4.31 | 5.63 | 8.94 | **12.75** | 40.00 |
| mamba2-1.3b | | | | | | |
| Base Model | 9.52 | 10.54 | 25.49 | 115.65 | 634.32 | 1479.45 |
| LongMamba | 10.12 | 10.31 | 11.36 | 11.61 | 12.81 | 13.55 |
| MambaExtend | 4.34 | 3.69 | 3.44 | 5.00 | 14.94 | 27.50 |
| Constant Scaling $A$ | 11.12 | 11.83 | 12.47 | 12.71 | 12.85 | 13.22 |
| Calibrated Scaling $A$ | 4.38 | 3.78 | **3.44** | **3.28** | **4.03** | **4.72** |

methods are fairly inconsistent on this front. Meanwhile, a constant scaling is generally ineffective, confirming previous doubts from Section 5.1 regarding the usefulness of a single constant factor based on the previous eigenvalue analysis. These results further support our analysis regarding the use of scaling factors for $A$ for length generalization compared to a wide variety of methods.

# 7 Conclusion

In this work, we conduct an in-depth exploration regarding the state transition matrix of `Mamba` models. We first provide a broader understanding of the SSM parameterization and how it can affect length generalization in `Mamba` models. In particular, we analyze the eigenvalue spectrum of both `Mamba` and `Mamba2` models, identifying the specific role this can have on the convergence of SSMs given long inputs. Then we identify how the scaling of $A$ as opposed to the more common practice of scaling $\Delta$ can be more effective at tuning this spectrum, enabling models to better generalize to long-contexts that far exceed the training context. We experiment on multiple long-context generalization tasks to validate that this newly built intuition holds empirically, on both `Mamba` and `Mamba2` models, highlighting the potential benefits of using $A$ for length generalization.

# 8 Acknowledgements

Jerry Huang was supported by the NSERC Canada Graduate Scholarships  Doctoral (CGS-D) program (funding reference number 589326) as well as the Bourse d'Éxcellence Hydro-Québec program.

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

# A    Proofs

## A.1    Proof of Lemma 4.1

**Lemma 4.1.** Let $\boldsymbol{B} \in \mathbb{R}^{d \times d}$ be a matrix with $\|\boldsymbol{B}\|_2 = \sigma_B$, and let $\boldsymbol{x} \in \mathbb{R}^d$ be a vector such that each entry of $\boldsymbol{x}$ satisfies $|x_i| \leq \sigma_x$. The upper bound for $\|\boldsymbol{B}\boldsymbol{x}\|_2$ is:

$$\|\boldsymbol{B}\boldsymbol{x}\|_2 \leq \sigma_B \cdot \sigma_x \cdot \sqrt{d}.$$

*Proof.* For any vector $\boldsymbol{x} \in \mathbb{R}^d$, it follows that:

$$\|\boldsymbol{B}\boldsymbol{x}\|_2 \leq \|\boldsymbol{B}\|_2 \cdot \|\boldsymbol{x}\|_2 .$$

Substituting $\|\boldsymbol{B}\|_2 = \sigma_B$, we obtain:

$$\|\boldsymbol{B}\boldsymbol{x}\|_2 \leq \sigma_B \cdot \|\boldsymbol{x}\|_2 .$$

And

$$\|\boldsymbol{x}\|_2 \leq \sqrt{\sum_{i=1}^{d} \sigma_x^2} = \sqrt{d} \cdot \sigma_x.$$

Substituting the bound on $\|\boldsymbol{x}\|_2$ into the inequality for $\|\boldsymbol{B}\boldsymbol{x}\|_2$, we have the norm of logit vector $u \in \mathbb{R}^d$:

$$\|\boldsymbol{u}\|_2 = \|\boldsymbol{B}\boldsymbol{x}\|_2 \leq \sigma_B \cdot \|\boldsymbol{x}\|_2 \leq \sigma_B \cdot \sqrt{d} \cdot \sigma_x.$$

$\square$

## A.2    Proof of Theorem 4.2

**Theorem 4.2.** Assume the transition matrix $\boldsymbol{\Lambda}$ is diagonal with eigenvalues $\lambda_i \sim \text{Uniform}[\lambda_{\min}, \lambda_{\max}]$ for $0 < \lambda_{\min} < \lambda_{\max} < 1$. Suppose the system evolves as

$$\boldsymbol{h}_t = \boldsymbol{\Lambda}\boldsymbol{h}_{t-1} + \boldsymbol{B}\boldsymbol{x}_t, \tag{10}$$

where $\boldsymbol{x}_t \sim \mathcal{N}(0, I)$ and $\boldsymbol{B}$ is a weight matrix whose rows are independently sampled as $\boldsymbol{b} \sim \mathcal{N}(0, \frac{1}{\sqrt{d}}\boldsymbol{I})$. Then, in the limit $t \to \infty$, the expected squared norm of the hidden state converges to

$$\mathbb{E}[\|\boldsymbol{h}_\infty\|^2] = \frac{1}{2(\lambda_{\max} - \lambda_{\min})} \log \left( \frac{1 - \lambda_{\min}^2}{1 - \lambda_{\max}^2} \right) \cdot \mathbb{E}[\|\boldsymbol{B}\boldsymbol{x}\|^2]. \tag{11}$$

*Proof.* We begin by unrolling the recurrence:

$$\boldsymbol{h}_t = \sum_{i=0}^{t-1} \boldsymbol{\Lambda}^i \boldsymbol{B}\boldsymbol{x}_{t-i}. \tag{12}$$

Assuming stationarity and independence of the inputs $\boldsymbol{x}_t$, the expected squared norm at steady state is

$$\mathbb{E}[\|\boldsymbol{h}_\infty\|^2] = \sum_{i=0}^{\infty} \mathbb{E}[\|\boldsymbol{\Lambda}^i \boldsymbol{B}\boldsymbol{x}\|^2]. \tag{13}$$

Consider the case of a single unit with eigenvalue $\lambda \in [\lambda_{\min}, \lambda_{\max}]$. The contribution of this unit is:

$$\mathbb{E}[h^2] = \sum_{i=0}^{\infty} \lambda^{2i} \mathbb{E}[\|\boldsymbol{b}\boldsymbol{x}\|^2] = \frac{\sigma^2}{1 - \lambda^2}, \tag{14}$$

where $\mathbb{E}[\|\boldsymbol{b}\boldsymbol{x}\|^2] = \mathbb{E}_b[\mathbb{E}_x[(bu)^2|b]] = \mathbb{E}_b[\|b\|^2] = \sigma^2$ is the contribution from the corresponding row of $\boldsymbol{B}$, and $\boldsymbol{B}$ is a weight matrix whose rows are independently sampled as $\boldsymbol{b} \sim \mathcal{N}(0, \frac{1}{\sqrt{d}}\boldsymbol{I})$.

With $\lambda \sim \mathrm{Uniform}[\lambda_{\min}, \lambda_{\max}]$, where $0 \leq \lambda_{\min} < \lambda_{\max} < 1$, the expected contribution over all units is

$$\mathbb{E}[\|\boldsymbol{h}_\infty\|^2] = d \cdot \mathbb{E}_\lambda \left[ \frac{\sigma^2}{1 - \lambda^2} \right] = \sigma^2 d \cdot \frac{1}{\lambda_{\max} - \lambda_{\min}} \int_{\lambda_{\min}}^{\lambda_{\max}} \frac{1}{1 - \lambda^2} \, d\lambda. \tag{15}$$

Evaluating the integral:

$$\int_{\lambda_{\min}}^{\lambda_{\max}} \frac{1}{1 - \lambda^2} \, d\lambda = \frac{1}{2} \log \left( \frac{1 - \lambda_{\min}^2}{1 - \lambda_{\max}^2} \right). \tag{16}$$

Hence,

$$\mathbb{E}[\|\boldsymbol{h}_\infty\|^2] = \sigma^2 d \cdot \frac{1}{2(\lambda_{\max} - \lambda_{\min})} \log \left( \frac{1 - \lambda_{\min}^2}{1 - \lambda_{\max}^2} \right). \tag{17}$$

Since $\mathbb{E}[\|\boldsymbol{Bx}\|^2] = d \cdot \sigma^2$, we obtain the final expression:

$$\mathbb{E}[\|\boldsymbol{h}_\infty\|^2] = \frac{1}{2(\lambda_{\max} - \lambda_{\min})} \log \left( \frac{1 - \lambda_{\min}^2}{1 - \lambda_{\max}^2} \right) \cdot \mathbb{E}[\|\boldsymbol{Bx}\|^2]. \tag{18}$$

$\square$

## A.3  Proof of Corollary 4.3 and Corollary 4.4

**Corollary 4.3** [Norm of `Mamba` State] Suppose the diagonal entries of $\boldsymbol{\Lambda}$ are independently drawn from a uniform distribution on $[0, \lambda]$, a moderate discretized step value $\Delta$ and the system evolves as $\boldsymbol{h}_t = \boldsymbol{\Lambda} \boldsymbol{h}_{t-1} + \overline{\boldsymbol{B}} \boldsymbol{x}_t = \mathrm{diag}\left( \exp(-\Delta \alpha) \right) \boldsymbol{h}_{t-1} + \Delta \boldsymbol{B} \boldsymbol{x}_t$. Then the convergence rate $\rho$ of the expected squared norm of the limiting state satisfies $\mathcal{O}\left( \frac{\Delta}{2\lambda} \log \left( \frac{1}{1 - \lambda^2} \right) \right)$.

*Proof.* Given Theorem 4.2, the convergence rate $\rho$ of `Mamba` state can be estimated as $\lambda_{min} \to 0$:

$$\rho = \lim_{t \to \infty} \frac{\mathbb{E}[\|\boldsymbol{h}_t\|^2]}{\mathbb{E}[\|\overline{\boldsymbol{B}} \boldsymbol{x}\|^2]} = \lim_{t \to \infty} \frac{\Delta \mathbb{E}[\|\boldsymbol{h}_t\|^2]}{\mathbb{E}[\|\boldsymbol{Bx}\|^2]} = \frac{\Delta}{2\lambda} \log \left( \frac{1}{1 - \lambda^2} \right) \tag{19}$$

$\square$

**Corollary 4.4** [Norm of `Mamba2` State]  Suppose $\boldsymbol{\Lambda} = \lambda \boldsymbol{I} = \exp(-\Delta \alpha) \boldsymbol{I}$ is a scalar multiple of the identity matrix, where $\lambda \in (0, 1)$, a moderate discretized step value $\Delta$ and the system evolves as $\boldsymbol{h}_t = \boldsymbol{\Lambda} \boldsymbol{h}_{t-1} + \overline{\boldsymbol{B}} \boldsymbol{x}_t = \exp(-\Delta \alpha) \odot \boldsymbol{h}_{t-1} + \Delta \boldsymbol{B} \boldsymbol{x}_t$. Then the convergence rate $\rho$ of the expected squared norm of the limiting state can be estimated as $\mathcal{O}(\frac{\Delta \cdot \lambda}{1 - \lambda})$.

*Proof.* Given Theorem 4.2, the convergence rate $\rho$ of `Mamba2` state can be estimated as $\delta = |\lambda_{max} - \lambda_{min}| \to 0$:

$$\rho = \lim_{\substack{t \to \infty \\ \delta \to 0}} \frac{\mathbb{E}[\|\boldsymbol{h}_t\|^2]}{\mathbb{E}[\|\overline{\boldsymbol{B}} \boldsymbol{x}\|^2]} = \lim_{\substack{t \to \infty \\ \delta \to 0}} \frac{\Delta \mathbb{E}[\|\boldsymbol{h}_t\|^2]}{\mathbb{E}[\|\boldsymbol{Bx}\|^2]} = \lim_{\delta \to 0} \frac{\Delta}{2\delta} \log \left( \frac{1 - \lambda_{min}^2}{1 - (\lambda_{min} + \delta)^2} \right) \tag{20}$$

Let $\lambda_{\min} = \lambda, \quad \lambda_{\max} = \lambda + \delta, \quad \delta \to 0$

Substitute into the expression:

$$\rho = \lim_{\delta \to 0} \frac{\Delta}{2\delta} \log \left( \frac{1 - \lambda^2}{1 - (\lambda + \delta)^2} \right) = \lim_{\delta \to 0} \frac{\Delta}{2\delta} \log \left( \frac{1 - \lambda^2}{1 - \lambda^2 - 2\lambda\delta - \delta^2} \right) \tag{21}$$

Let $\lambda = \lambda_{\min}, \delta = \lambda_{\max} - \lambda$, then:

$$= \lim_{\delta \to 0} \frac{\Delta}{2\delta} \log \left( \frac{1 - \lambda^2}{1 - (\lambda + \delta)^2} \right) \tag{22}$$

$$= \lim_{\delta \to 0} \frac{\Delta}{2\delta} \log \left( \frac{1 - \lambda^2}{1 - \lambda^2 - 2\lambda\delta - \delta^2} \right) \tag{23}$$

$$= \lim_{\delta \to 0} \frac{\Delta}{2\delta} \log \left( 1 + \frac{2\lambda\delta + \delta^2}{1 - \lambda^2} \right) \tag{24}$$

$$\approx \lim_{\delta \to 0} \frac{\Delta}{2\delta} \cdot \frac{2\lambda\delta + \delta^2}{1 - \lambda^2} \tag{25}$$

$$= \frac{\Delta\lambda}{1 - \lambda^2} \tag{26}$$

$\square$

## B   Elaboration on Theorem 4.2

The divergent convergence behavior is irrespective of spectral assumptions. We provide an elaboration on Theorem 4.2 and corresponding convergence analysis without imposing any strict distribution assumptions.

### B.1   Setup and notation

Let $d, m \in \mathbb{N}$. For $i = 1, \ldots, d$ let $\lambda_i \in (0, 1)$ be i.i.d. samples from a density $p(\lambda)$ supported on $[\lambda_{\min}, \lambda_{\max}] \subset (0, 1)$. Denote

$$\Lambda = \mathrm{diag}(\lambda_1, \ldots, \lambda_d) \in \mathbb{R}^{d \times d}.$$

Consider the linear system

$$\mathbf{h}_t = \Lambda \mathbf{h}_{t-1} + \mathbf{B} \mathbf{x}_t, \qquad t \geq 1, \tag{27}$$

where $\{\mathbf{x}_t\}$ are i.i.d. $\mathcal{N}(0, I_m)$ and $\mathbf{B} \in \mathbb{R}^{d \times m}$ has i.i.d. rows $\mathbf{b}_1, \ldots, \mathbf{b}_d$, each distributed as

$$\mathbf{b}_i \sim \mathcal{N}\big(0, \Sigma_B\big),$$

with $\Sigma_B \in \mathbb{R}^{m \times m}$ a (given) covariance matrix.

Assume $\mathbf{h}_0 = 0$ (or any initial condition that decays under $\Lambda$). We are interested in the steady-state expected squared norm

$$\mathbb{E}\big[\|\mathbf{h}_\infty\|^2\big] := \lim_{t \to \infty} \mathbb{E}\big[\|\mathbf{h}_t\|^2\big],$$

where the expectation is over the driving noise $\{\mathbf{x}_t\}$ and the random matrix $\mathbf{B}$ and the random eigenvalues $\{\lambda_i\}$.

### B.2   Theorem (Expected state norm under general spectral distribution)

**Theorem B.1.** *Under the assumptions above, the limit $\mathbb{E}[\|\mathbf{h}_\infty\|^2]$ exists and equals*

$$\boxed{\mathbb{E}\big[\|\mathbf{h}_\infty\|^2\big] = \mathbb{E}\big[\|\mathbf{B}\mathbf{x}\|^2\big] \cdot \int_{\lambda_{\min}}^{\lambda_{\max}} \frac{p(\lambda)}{1 - \lambda^2} \, d\lambda,} \tag{28}$$

*where $\mathbf{x} \sim \mathcal{N}(0, I_m)$ is independent of $\mathbf{B}$ and $\lambda$, and the expectation on the left of the product is taken over $\mathbf{B}$ and $\mathbf{x}$.*

*Proof.* Because $\Lambda$ is diagonal and $\mathbf{x}_t \sim \mathcal{N}(0, I_m)$ i.i.d., the process equation 27 is Gaussian with zero mean for all $t$. Let

$$\Sigma_t := \mathbb{E}[\mathbf{h}_t \mathbf{h}_t^\top] \in \mathbb{R}^{d \times d}$$

be the (time-$t$) covariance of the hidden state. From equation 27 we have the Lyapunov-type recursion [108, 2]

$$\Sigma_t = \Lambda \Sigma_{t-1} \Lambda + \mathbb{E}[\mathbf{B}\mathbf{x}_t \mathbf{x}_t^\top \mathbf{B}^\top].$$

Since $\mathbf{x}_t\mathbf{x}_t^\top$ has expectation $I_m$ and is independent of $\mathbf{B}$ and $\Lambda$, the driving covariance is
$$Q := \mathbb{E}[\mathbf{B}\mathbf{x}_t\mathbf{x}_t^\top\mathbf{B}^\top] = \mathbb{E}[\mathbf{B}\mathbf{B}^\top].$$
Because $\Lambda$ is diagonal, the steady-state covariance $\Sigma_\infty := \lim_{t\to\infty}\Sigma_t$ is also diagonal; denote its diagonal entries by $s_i := (\Sigma_\infty)_{ii}$, $i = 1,\ldots,d$. The scalar recurrence for each diagonal entry is
$$s_i = \lambda_i^2 s_i + Q_{ii},$$
hence (since $|\lambda_i| < 1$)
$$s_i = \frac{Q_{ii}}{1 - \lambda_i^2}.$$
Therefore the steady-state expected squared norm equals
$$\mathbb{E}\big[\|\mathbf{h}_\infty\|^2\big] = \mathbb{E}\big[\mathrm{trace}(\Sigma_\infty)\big] = \mathbb{E}\left[\sum_{i=1}^d \frac{Q_{ii}}{1 - \lambda_i^2}\right].$$

Because the pairs $(Q_{ii}, \lambda_i)$ are i.i.d. across $i$ and rows of $\mathbf{B}$ are independent of the $\lambda_i$'s, we have for a generic row $\mathbf{b} \in \mathbb{R}^m$
$$\mathbb{E}\big[\|\mathbf{h}_\infty\|^2\big] = d\,\mathbb{E}\left[\frac{\|\mathbf{b}\|^2}{1 - \lambda^2}\right] = d\,\mathbb{E}[\|\mathbf{b}\|^2]\,\mathbb{E}\left[\frac{1}{1 - \lambda^2}\right],$$
where $\lambda$ is a generic draw from $p(\lambda)$ and independence between $\mathbf{b}$ and $\lambda$ was used to factor the expectation.

Observe that
$$\mathbb{E}\big[\|\mathbf{B}\mathbf{x}\|^2\big] = \mathbb{E}_\mathbf{B}\mathbb{E}_\mathbf{x}\big[\|\mathbf{B}\mathbf{x}\|^2\big] = \mathbb{E}_\mathbf{B}\big[\mathrm{trace}(\mathbf{B}^\top\mathbf{B})\big] = \mathbb{E}_\mathbf{B}\Big[\sum_{i=1}^d \|\mathbf{b}_i\|^2\Big] = d\,\mathbb{E}[\|\mathbf{b}\|^2].$$

Combining the last two displayed equalities yields equation 28:
$$\mathbb{E}\big[\|\mathbf{h}_\infty\|^2\big] = \mathbb{E}\big[\|\mathbf{B}\mathbf{x}\|^2\big]\,\mathbb{E}\left[\frac{1}{1 - \lambda^2}\right] = \mathbb{E}\big[\|\mathbf{B}\mathbf{x}\|^2\big]\int_{\lambda_{\min}}^{\lambda_{\max}} \frac{p(\lambda)}{1 - \lambda^2}\,d\lambda.$$
This completes the proof. $\qquad\square$

## B.3 Evaluation of $\mathbb{E}\big[\|\mathbf{B}\mathbf{x}\|^2\big]$ in the isotropic row case

If each row $\mathbf{b}_i \sim \mathcal{N}(0, \sigma_B^2 I_m)$ (i.i.d. across rows), then
$$\mathbb{E}[\|\mathbf{b}\|^2] = \mathrm{trace}(\sigma_B^2 I_m) = m\sigma_B^2, \qquad \mathbb{E}\big[\|\mathbf{B}\mathbf{x}\|^2\big] = d\,m\,\sigma_B^2.$$
In the special (informal) normalization used in the statement above, where each row has covariance $\Sigma_B = (1/\sqrt{d})I_m$, one has $\sigma_B^2 = 1/\sqrt{d}$ and hence
$$\mathbb{E}\big[\|\mathbf{B}\mathbf{x}\|^2\big] = dm\frac{1}{\sqrt{d}} = md^{1/2}.$$

## B.4 Asymptotic analysis of the integral

Define
$$I(\lambda_{\min}, \lambda_{\max}) := \int_{\lambda_{\min}}^{\lambda_{\max}} \frac{p(\lambda)}{1 - \lambda^2}\,d\lambda = \mathbb{E}\left[\frac{1}{1 - \lambda^2}\right].$$

**(1) As $\lambda_{\max} \to 1^-$.** Near $\lambda = 1$ we have the expansion $1 - \lambda^2 = (1 - \lambda)(1 + \lambda) \approx 2(1 - \lambda)$. Suppose $p$ is continuous at $\lambda = 1$ and $p(1) > 0$. For $\lambda$ close to 1,
$$\frac{p(\lambda)}{1 - \lambda^2} \sim \frac{p(1)}{2} \cdot \frac{1}{1 - \lambda}.$$
Hence for $\lambda_{\max}$ sufficiently close to 1,
$$I(\lambda_{\min}, \lambda_{\max}) = \int_{\lambda_{\min}}^{\lambda_{\max}} \frac{p(\lambda)}{1 - \lambda^2}\,d\lambda \approx \frac{p(1)}{2}\int_{\lambda_{\min}}^{\lambda_{\max}} \frac{1}{1 - \lambda}\,d\lambda = \frac{p(1)}{2}\log\left(\frac{1 - \lambda_{\min}}{1 - \lambda_{\max}}\right).$$
Therefore
$$I(\lambda_{\min}, \lambda_{\max}) \sim -\frac{p(1)}{2}\log(1 - \lambda_{\max}) \quad \text{as } \lambda_{\max} \to 1^-,$$
and in particular $I(\lambda_{\min}, \lambda_{\max}) \to +\infty$ with logarithmic divergence.

**(2) As $\lambda_{\max} \to 0^+$.**  When $\lambda_{\max}$ is small, $1 - \lambda^2 \approx 1$, so the integrand is approximately $p(\lambda)$. If $p$ is continuous near 0 with $p(0) > 0$, then

$$I(\lambda_{\min}, \lambda_{\max}) \approx \int_{\lambda_{\min}}^{\lambda_{\max}} p(\lambda) \, d\lambda \approx p(0) \cdot (\lambda_{\max} - \lambda_{\min}).$$

If we also take $\lambda_{\min} = 0$ or consider the leading-order scaling in $\lambda_{\max}$, then

$$I(\lambda_{\min}, \lambda_{\max}) \sim p(0) \, \lambda_{\max} \quad \text{as } \lambda_{\max} \to 0^+,$$

i.e., $I$ vanishes linearly with $\lambda_{\max}$.

Combining the prefactor $\mathbb{E}[\|\mathbf{Bx}\|^2]$ from equation 28 with the asymptotics above yields

$$\boxed{\mathbb{E}\big[\|\mathbf{h}_\infty\|^2\big] \sim \begin{cases} -\dfrac{p(1)}{2} \, \mathbb{E}\big[\|\mathbf{Bx}\|^2\big] \, \log(1 - \lambda_{\max}), & \lambda_{\max} \to 1^-, \\ p(0) \, \mathbb{E}\big[\|\mathbf{Bx}\|^2\big] \, \lambda_{\max}, & \lambda_{\max} \to 0^+. \end{cases}}$$

# C  Additional Experimental Details and Results

## C.1  Technical Details

All experiments were conducted on a single machine with 2 NVIDIA RTX4080 16GB GPUs. Experiments were run in an environment using CUDA version 12.6 and PyTorch 2.6.0.

## C.2  Constant Scaling Language Modeling Perplexity

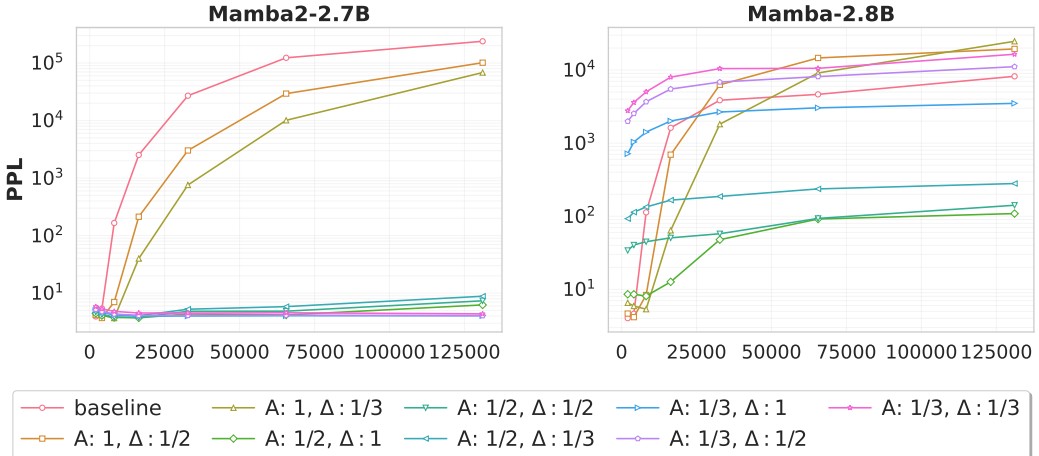

Figure 5: Language modeling perplexity on ProofPile after applying a constant scaling factor to either $A$ or $\Delta_t$. The red line with 'o' mark indicates the baseline, where neither $A$ nor $\Delta_t$ are scaled.

### C.3 `MambaExtend` Calibration

Here, we give an overview of the calibration functions we use within our `MambaExtend`-based experiments. Each of the described methods replace the calibration function CF within Algorithm 1. In our explicit implementation for calibrating scaling factors for $\boldsymbol{A}$, we use the same hyperparameters as Azizi et al. [3].

**Calibration via back-propagation.**   To train the un-frozen calibration parameters on a calibration set, we apply a back-propagation algorithm to find the optimal scaling factors. This is described in Algorithm 2.

---

**Algorithm 2** Calibration via back-propagation

---

1: **Input**: Frozen model $\mathcal{M}$, calibration set $\mathcal{C}$, initial scaling factors $\boldsymbol{S}$. Learning rate $\eta$, perturbation magnitude $c$, iterations $K$
2: **Output**: Learned scaling factors $\boldsymbol{S} = [\boldsymbol{s}_1, \ldots, \boldsymbol{s}_L] \in \mathbb{R}_+^{d_s \times L}$
3: optimizer $= \mathrm{Adam}(\boldsymbol{S}, \eta)$
4: **for** $k \leq K$ **do**
5:     $\mathcal{L} = \mathsf{eval}\,(\mathcal{M}_{c \times \boldsymbol{S}^+}, \mathcal{C})$
6:     $\mathcal{L}.\mathsf{backward}\,()$
7:     optimizer.step $()$
8:     $\boldsymbol{S} \leftarrow \mathsf{clamp}\,(\boldsymbol{S}, 0.001)$
9: **end for**
10: **return** $\boldsymbol{S}$

---

**Calibration via zeroth-order optimization.**   Zeroth-order optimization offers an efficient yet noisier method for calibration, as it relies solely on forward passes to approximate gradients. Specifically, this is a multi-iteration process in which, at each iteration, the scaling factors are randomly perturbed using a random variable $\delta$ sampled from a Rademacher distribution. The magnitude of the perturbation and the learning rate for the updates are controlled by the hyper-parameters $c$ and $\eta$, respectively. We employ the two-sided variant of the simultaneous perturbation stochastic approximation method (SPSA) [110], which obtains gradient approximations by applying both positive and negative perturbations to the parameters simultaneously. The two-sided SPSA approach yields gradient estimates with lower variance than the one-sided version, thus enhancing accuracy, especially in noisy environments [119]. This is described in Algorithm 3.

---

**Algorithm 3** Calibration via zeroth-order optimization

---

1: **Input**: Frozen model $\mathcal{M}$, calibration set $\mathcal{C}$, perturbation magnitude $c$, iterations $K$
2: **Output**: Learned scaling factors $\boldsymbol{S} = [\boldsymbol{s}_1, \ldots, \boldsymbol{s}_L] \in \mathbb{R}_+^{d_s \times L}$
3: **for** $k \leq K$ **do**
4:     $\delta \in \mathbb{R}^{d_s \times L} \sim \mathrm{Radamacher}()$
5:     $\boldsymbol{S}^+ = \boldsymbol{S} + c \times \delta$
6:     $\boldsymbol{S}^- = \boldsymbol{S} - c \times \delta$
7:     $\ell^+ = \mathsf{eval}(\mathcal{M}_{c \times \boldsymbol{S}^+}, \mathcal{C})$
8:     $\ell^- = \mathsf{eval}(\mathcal{M}_{c \times \boldsymbol{S}^-}, \mathcal{C})$
9:     $\hat{\nabla}_{\boldsymbol{S}} = (\ell^+ - \ell^-)/(2 \cdot c \cdot \delta)$
10:     $\boldsymbol{S} \leftarrow \boldsymbol{S} - \eta \cdot \hat{\nabla}_{\boldsymbol{S}}$
11:     $\boldsymbol{S} \leftarrow \mathsf{clamp}\,(\boldsymbol{S}, 0.001)$
12: **end for**
13: **return** $\boldsymbol{S}$

---

### C.3.1 Language Modeling Perplexity

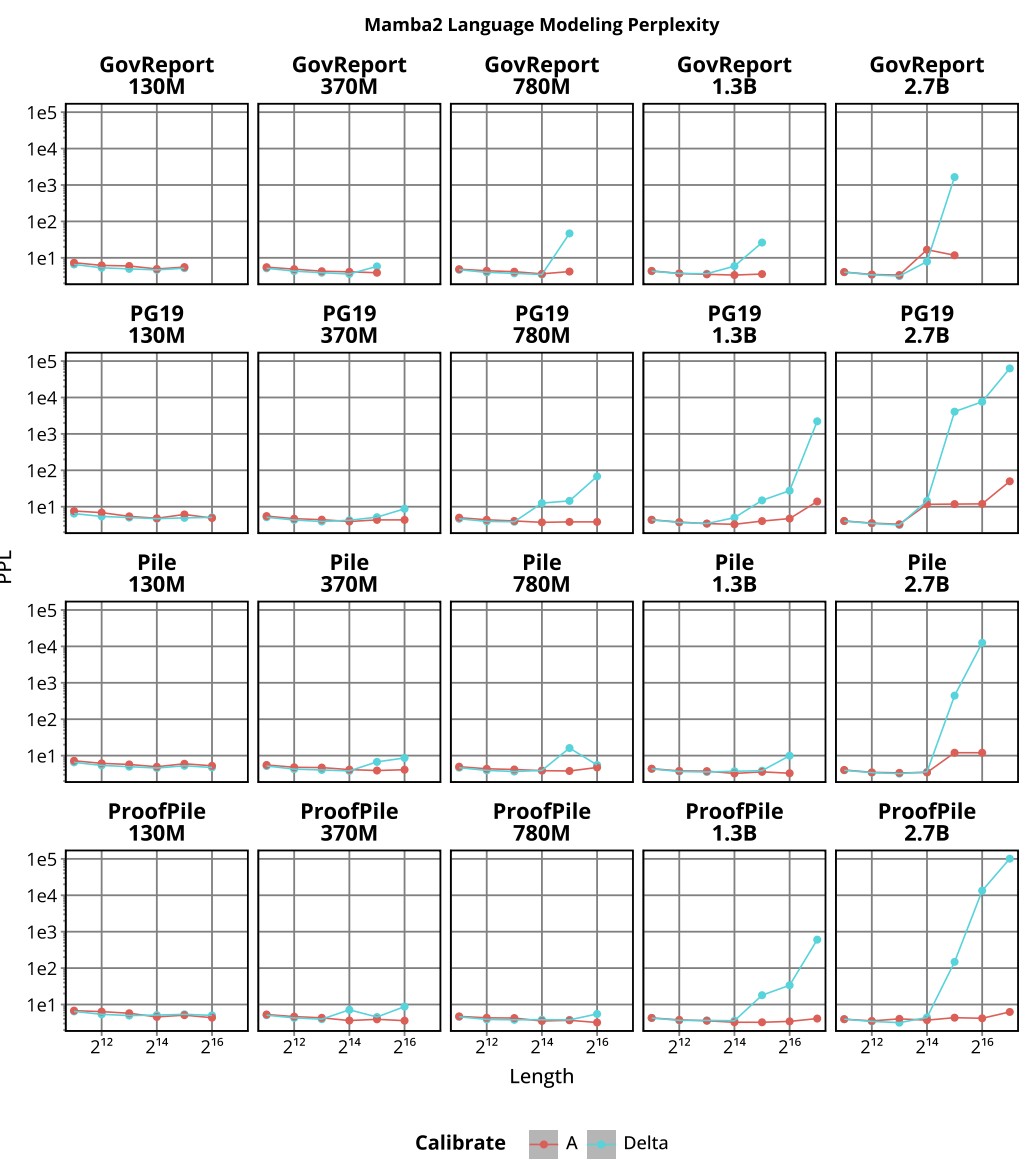

Figure 6: Language Model Perplexity performance of `Mamba2` models by calibrating scaling factors for either $\log(A)$ (red lines) or $\Delta_t$ (cyan lines). Perplexities are reported across various datasets (GovReport, PG19, ProofPile, Pile) as well as model sizes.

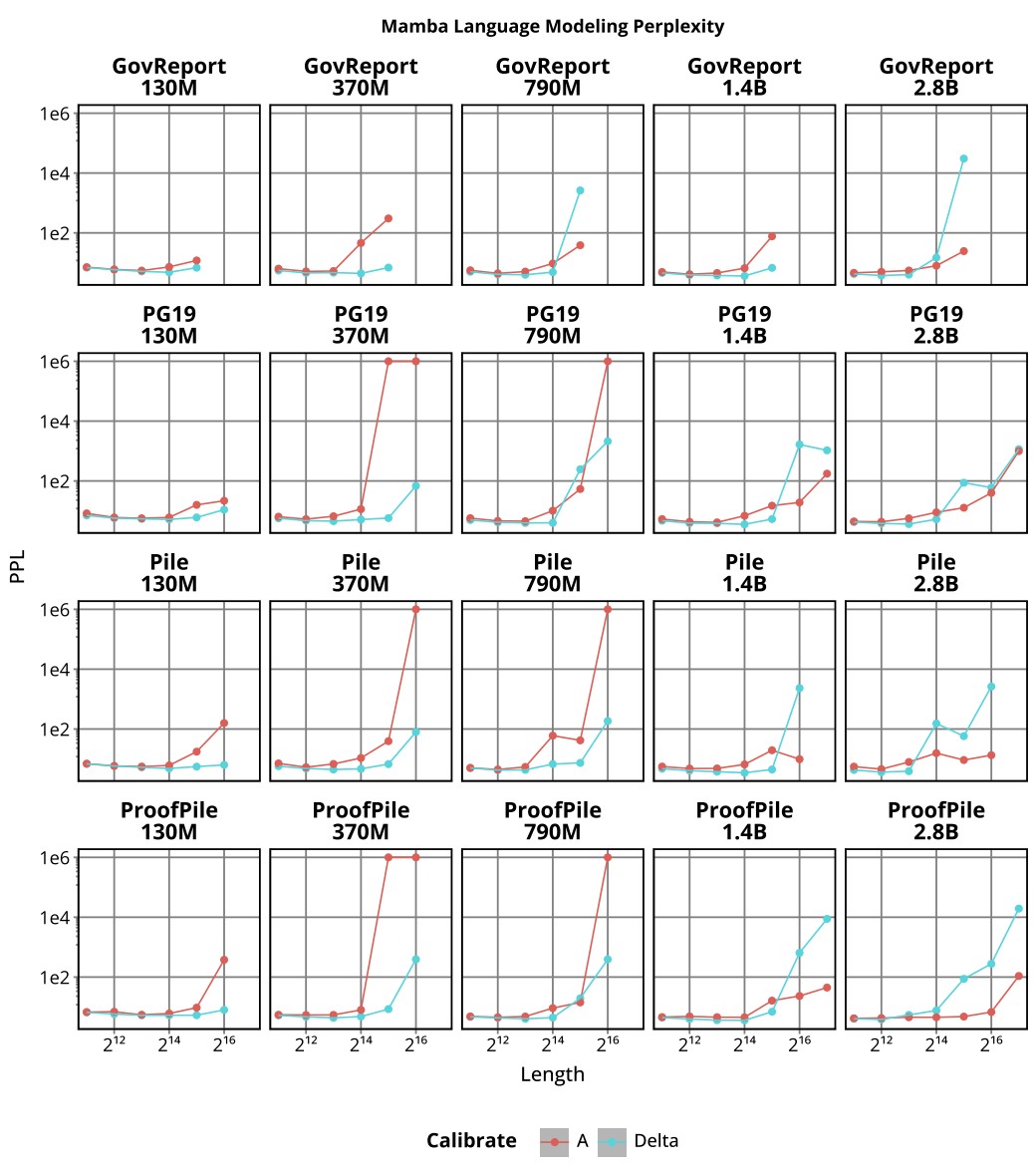

Figure 7: Language Model Perplexity performance of `Mamba` models by calibrating scaling factors for either $\log(A)$ (red lines) or $\Delta_t$ (cyan lines). Perplexities are reported across various datasets (GovReport, PG19, ProofPile, Pile) as well as model sizes.

## C.3.2 Passkey Retrieval

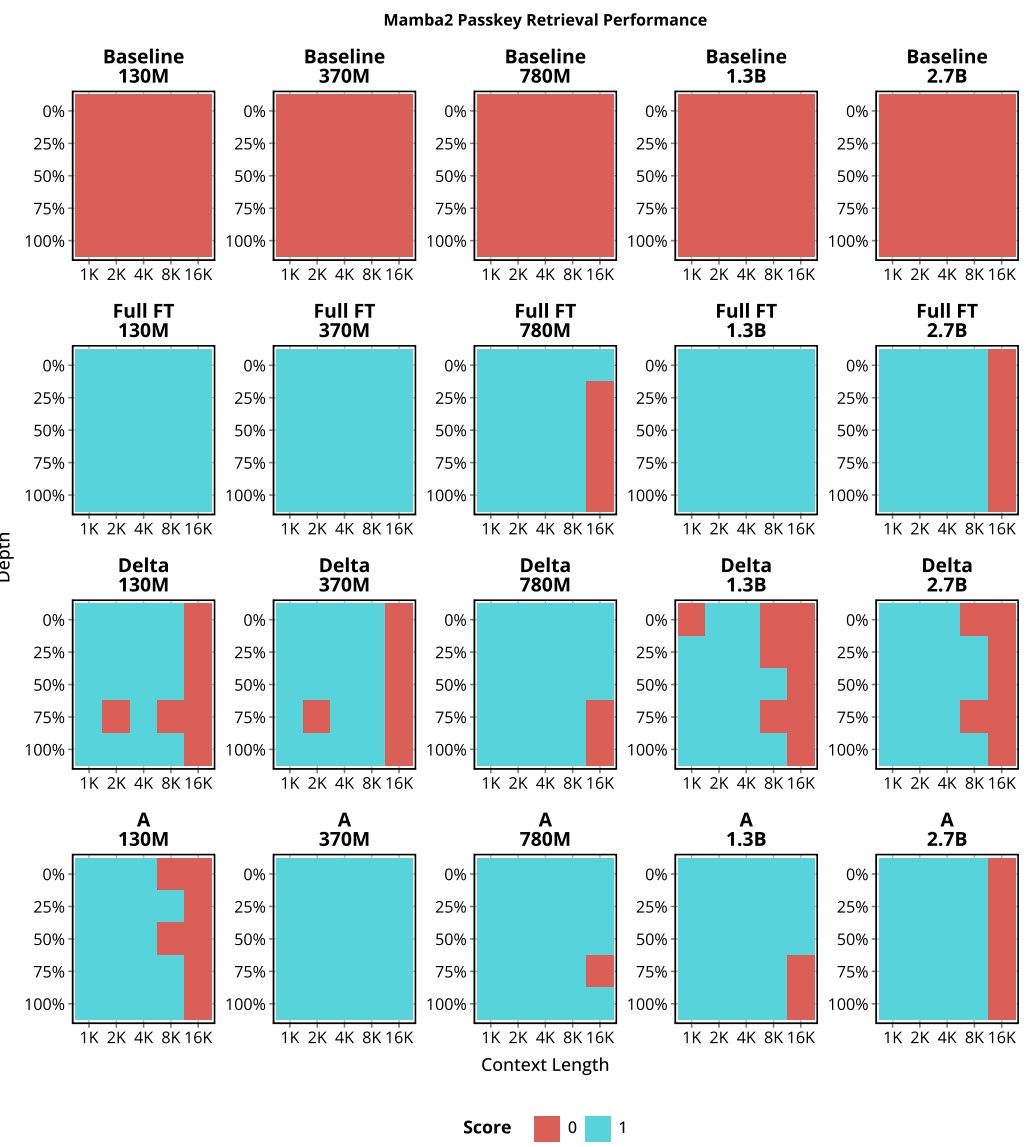

Figure 8: Passkey Retrieval performance of `Mamba2` models by calibrating scaling factors for either $\log(\boldsymbol{A})$ or $\Delta_t$. Blue squares mean that the model was able to solve all examples of the given evaluation length/depth pair after tuning scaling factors, while red squares means that at least one mistake was made, i.e. an incorrect passage was retrieved.

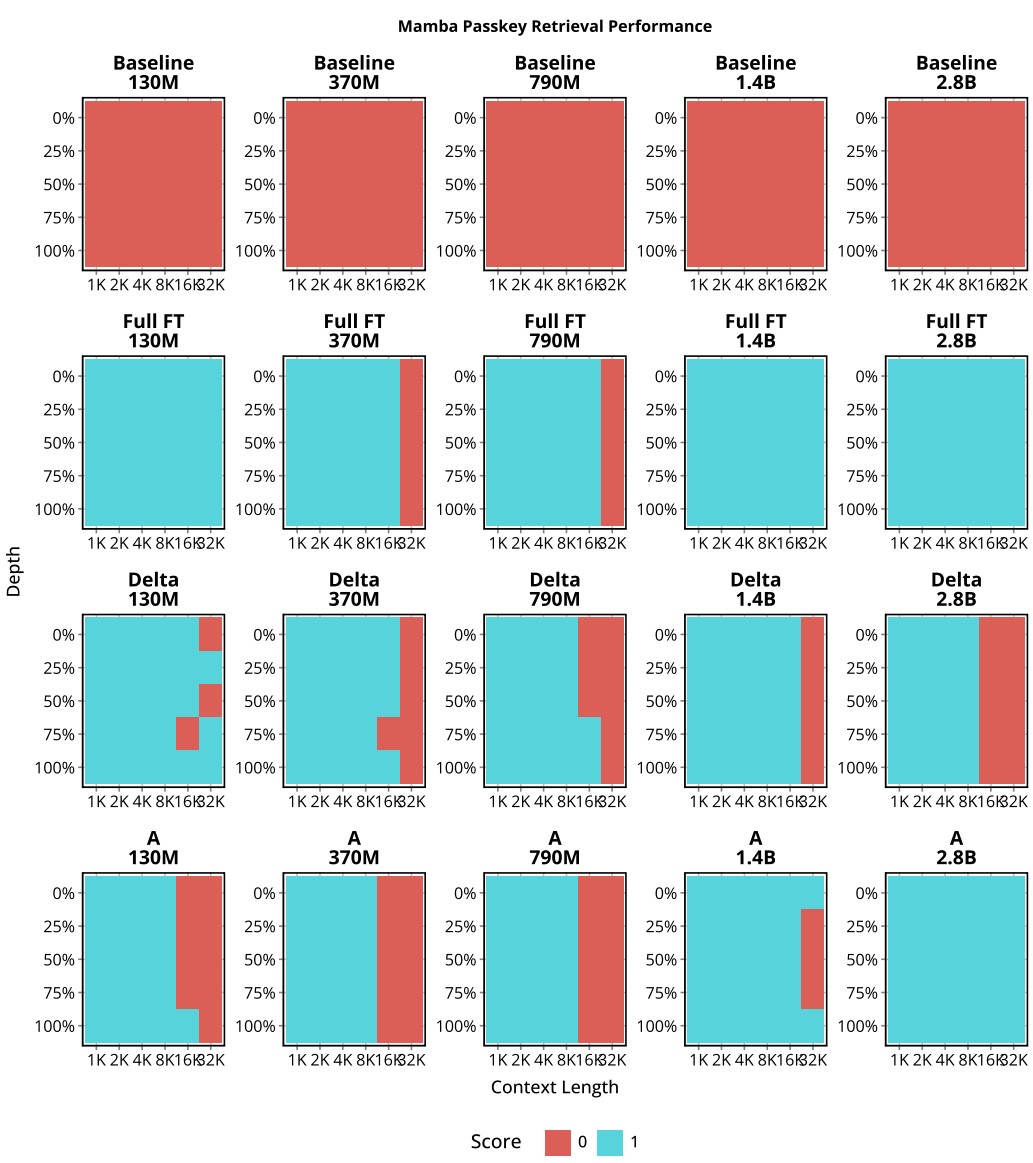

Figure 9: Passkey Retrieval performance of `Mamba` models by calibrating scaling factors for either $\log(A)$ or $\Delta_t$. Blue squares mean that the model was able to solve all examples of the given evaluation length/depth pair after tuning scaling factors, while red squares means that at least one mistake was made, i.e. an incorrect passage was retrieved.

### C.3.3  LongBench

We evaluate the following tasks from `LongBench` (Table 4). Due to our pre-training on an English dataset, we choose to use only the English language tasks included in the benchmark.

Table 4: Tasks from LongBench on which we evaluate.

| Task | Context Type | Average Length | Metric | Data Samples |
|------|------|------|------|------|
| QASPERQA [23] | Science | 3619 | F1 | 200 |
| HOTPOTQA [145] | Wikipedia | 9151 | F1 | 200 |
| 2WIKIMULTIQA [54] | Wikipedia | 4887 | F1 | 200 |
| TREC [78] | Web Questions | 5117 | Accuracy | 200 |
| TRIVIAQA [70] | Wikipedia/Web | 8209 | F1 | 200 |
| LCC [50] | Github | 1235 | Edit Similarity | 500 |
| REPOBENCH-P [82] | Github Repositories | 4206 | Edit Similarity | 500 |

### C.4  Pre-Trained Model Checkpoints Used

We use the official pre-trained model checkpoints of `Mamba` from the Hugging Face model Hub, found at `https://huggingface.co/state-spaces.`:

- `state-spaces/mamba-130m`
- `state-spaces/mamba-370m`
- `state-spaces/mamba-790m`
- `state-spaces/mamba-1.4b`
- `state-spaces/mamba-2.8b`
- `state-spaces/mamba2-130m`
- `state-spaces/mamba2-370m`
- `state-spaces/mamba2-780m`
- `state-spaces/mamba2-1.3b`
- `state-spaces/mamba2-2.7b`

We use the original `Mamba`[5] implementations for adjusting scaling parameters.

---

[5]`https://github.com/state-spaces/mamba`

# D  Broader Impacts

This work explores a novel method for length generalization of `Mamba`-based language models. While the direct usage of such models can entail potential broader risks within AI-based systems if potentially trained to scale, these risks do not stem directly from the methods and analysis presented within the paper. As such, there are no risks that are deemed significant and worthy of further discussion.

# E  Limitations

The primary limitation of our current work is that it is focused on `Mamba`-style models; as such, the methodology requires adaptation to similar models that utilize state-transition dynamics. However, many analogies exist between how information is written to the state and read out from the memory, presenting a potential avenue for use in other such models.

Another potential limitation is the lack of instruction-tuned models that are available for direct use, limiting the set of experiments and evaluations on which can be adequately conducted.

