# OpenReview forum: "Mamba Modulation: On the Length Generalization of Mamba Models"
_NeurIPS.cc/2025/Conference — NeurIPS 2025 poster_

### Official Review · Reviewer_z8Ky · 2025-06-17

**Clarity:** 3
**Significance:** 2
**Originality:** 2
**Rating:** 3
**Confidence:** 4

**Summary:**

This paper explores the length generalization capabilities of Mamba. The paper provides theoretical arguments tying the spectrum of the transition matrix to the convergence behavior of the hidden state when applied to infinitely long sequences. The paper demonstrates that scaling the spectrum of the transition matrix directly is favorable compared to previous works that scale the discretization step.
Supporting experiments are provided.

**Questions:**

- The sentence in lines 30-33 asserts that the primary motivation for mamba is length generalization capabilities. Can you please provide grounds for this claim? To the best of my knowledge Mamba shows superior performance compared to previous SSMs on long sequences (not necessarily in the extrapolation setup).

- What is $s$ in line 151? it isn't defined before it is used.

- Figure 1 isn't clear from its description - Specifically, this is the spectrum per layer of a specific model. What does it mean that certain layers have fast decay while others decay slowly?

- Why is Lemma 4.1. state as a formal lemma? this is trivial and does not require a dedicated proof.

- In Theorem 4.2, Corollary 4.3 & Corollary 4.4 - why is the assumption on the eigenvalue distribution different per statement? In general the statements are not very clear when written verbally, I would suggest writing the statement formally (in lines 192-193 and 196-197).

**Ethical Concerns:**

["NO or VERY MINOR ethics concerns only"]

**Final Justification:**

The authors have conducted additional experiments. After reviewing other reviewers’ comments, I am persuaded that the paper’s significance is greater than I initially assessed.

**Limitations:**

The empirical benefit of scaling $\mathbf{A}$ instead of $\mathbf{\Delta}$ isn't clear. Results seem to be marginally better in certain setups and worse in others. The authors fail to explicitly discuss this in the paper. It would be good to discuss in a dedicated section the tradeoffs and benefits of the different scaling methods, as it stands the method presented seems to offer incremental novelty compared to MambaExtend.

**Paper Formatting Concerns:**

The paper has multiple flaws in the form of unclear sentences and other formatting issues, below are a few:
- Sentence in lines 36-37 isn't clear.
- Sentence in line 59 isn't clear - "Broadly, summarize our contributions as follows"
- Section 3.1 - Eq. (1) - the dimensions of $\mathbf{C}$ don't work, either use $\mathbf{C}^{\top}$ or swap the dimensions of $\mathbf{C}$.
- line 132 - $\alpha$ should be $\alpha_i \forall i\in \lbrace 1,\dots, d\rbrace$.
- Sentence in lines 132-133: "Mamba makes $\mathbf{\Delta, B}$ and $\mathbf{C}$ to be input dependent" doesn't seem to be proper English.
- line 137 - should it be $\mathbf{C}_t$ or $\bar{\mathbf{C}}_t$?
- Eq. (5) - $C_i$ should be in bold.
- line 145 - should the product be until $i$ instead of $L$? (same question for Eq. (7))
- Figure 2 legend isn't clear and does not align with the textual description - delta scale should be in red or black?

**Quality:**

3

**Strengths And Weaknesses:**

**Strengths**:
* The paper discusses a very relevant and vibrant topic of length generalization.
* The paper is written well and structured in an intuitive flow.
* The presented method for scaling the eigenvalues (and improving length generalization) is simple and clear.

**Weaknesses**:
* Results in the paper are trivial and the significance seems very limited.
* The paper does not discuss known results on other architectures or distinguishes the results and the novelty brought forth in the paper.
* Related work is very lacking - missing works on extrapolation and length generalization in different setups. I would expect a paper discussing length generalization of Mamba to discuss:
   - Works related to extrapolation in transformers:
      - Train Short, Test Long: Attention with Linear Biases Enables Input Length Extrapolation. Press et al. ICLR 2022.
      - Unveiling transformers with lego: a synthetic reasoning task. Zhang et al. Arxiv 2022.
      - etc.
   - Works discussing length generalization and extrapolation in RNNs:
      - Implicit Bias of Linear RNNs. Emami et al. ICML 2021.
      - On the implicit bias of gradient descent for temporal extrapolation. Cohen-Karlik et al. AISTATS 2022.
      - Learning Low Dimensional State Spaces with Overparameterized Recurrent Neural Network. Cohen-Karlik et al. ICLR 2023.
      - etc.

---

> ### Author Rebuttal · Authors · 2025-07-31
>
> First, we thank the reviewer for acknowledging the **relevant and vibrant topic** of our work, that it is **written well and structured** and the presented method is **simple and clear**. We also acknowledge the various points of concern and questions they have raised, which we hope will be resolved with the following response.
>
> -----
>
> > W1. Results in the paper are trivial and the significance seems very limited.
>
> **A1**: We understand these concerns, but highlight that our method offers **significant/consistent improvements** over the original pre-trained models.
>
> | PPL (**mamba-1.4b**)| 2k | 4k | 8k | 16k | 32k | 64k |
> |-|-|-|-|-|-|-|
> | Base | 9.67   | 10.23  | 11.43  | 17.46  | 59.77   | 444.09   |
> | Ours | **5.31** | **4.31** | **4.13** | **6.88** | **14.94** | **19.13** |
>
> | PPL (**mamba2-1.3b**)| 2k | 4k | 8k | 16k | 32k | 64k |
> |-|-|-|-|-|-|-|
> | Base | 9.52   | 10.54  | 25.49  | 115.65 | 634.32  | 1479.45  |
> | Ours | **4.38** | **3.78** | **3.44** | **3.28** | **4.03** | **4.72** |
>
> We posit that generalization to longer inputs remains a challenge for Mamba models. Our proposed scaling method demonstrates improvements under such conditions, offering evidence to support our theorems. This insight is also novel, both explaining why existing methods fall short in extrapolating and providing a principled criterion for designing more resilient architectures.
>
> > W2. The paper does not discuss known results on other architectures or distinguishes the results and the novelty brought forth in the paper.
>
> **A2**: Mamba models present a promising alternative to the well-established Transformer architecture. However, despite their rising popularity, they remain comparatively under-explored, particularly when it comes to understanding their limitations.
>
> We address this gap by uncovering the fundamental causes behind Mamba’s failure to generalize across sequence lengths, which previously has not been explained. Our analysis exposes key structural differences from self-attention mechanisms that contribute to this behaviour, providing valuable insight that deepens the field’s understanding and paves the way for more robust model design.
>
> > W3. Related work is very lacking - missing works on extrapolation and length generalization in different setups.
>
> **A**: We will add and discuss the mentioned references regarding length generalization in transformers and RNNs in the final version of our paper as follows.
>
> The literature on transformers highlights positional encoding and attention biases as critical for length extrapolation:
> Press et al. introduced ALiBi (Attention with Linear Biases), demonstrating that linear distance-based attention biases enable effective extrapolation beyond training lengths. Zhang et al. used synthetic reasoning tasks to analyze transformers' generalization capabilities, revealing how attention patterns and pretraining affect robustness to longer sequences.
> Emami et al. analyzed the implicit bias of linear RNNs in sequence modeling tasks. Cohen-Karlik et al. investigated gradient descent dynamics for temporal extrapolation and showed how overparameterized RNNs learn low-dimensional state spaces.
>
> While Mamba's efficiency is well-established, its extrapolation behavior requires systematic comparison to transformer approaches and integration of insights from RNN theoretical analyses.
>
>
> 1. Train Short, Test Long: Attention with Linear Biases Enables Input Length Extrapolation. Press et al. ICLR 2022.
> 2. Unveiling transformers with lego: a synthetic reasoning task. Zhang et al. Arxiv 2022.
> 3. Implicit Bias of Linear RNNs. Emami et al. ICML 2021.
> 4. On the implicit bias of gradient descent for temporal extrapolation. Cohen-Karlik et al. AISTATS 2022.
> 5. Learning Low Dimensional State Spaces with Overparameterized Recurrent Neural Nets. Cohen-Karlik et al. ICLR 2023.
>
> > Q1. Can you please provide grounds for [sentence in lines 30-33]?
>
> **A**: We acknowledge some potential confusion that may have stemmed from a poor choice of phrasing.
>
> The primary motivation of Mamba is with regards to the ability to have models address the computational inefficiency of models on long sequences, which is shown by results in the original paper. However, Mamba does claim extrapolation abilities, namely on synthetic tasks such as selective copying and induction heads, but the same experiments have not been tested on language data, which is why we believe this claim prompts further investigation.
>
> > Q2. What is s in line 151? it isn't defined before it is used.
>
> **A**: ‘s’ simply appears after its definition:  ‘scalar values’, refers to the scaling factors used by Azizi et al. to multiply with $\Delta_t$.
>
> > Q3. Figure 1 isn't clear from its description. What does it mean that certain layers have fast decay while others decay slowly?
>
> **A**: Figure 1 visualizes the spectrum of each layer to motivate our theoretical analysis. It reveals a consistent pattern across layers: the presence of both large (near 1) and small (near 0) eigenvalues. This is not incidental; it underpins Section 4.2, where the convergence behaviour is shown to depend critically on the spectral distribution. If the largest eigenvalues of different states in a layer are concentrated close to zero, then the information contained in the states of this layer decays fast. Conversely, layers with slower spectral decay (i.e., eigenvalues closer to 1) preserve state information for longer durations.
>
> > Q4. Lemma 4.1. is trivial and does not require a dedicated proof.
>
> **A**: This Lemma ensures that we do not need to impose any explicit assumption on the norm of the input. By presenting it as a lemma, we establish a clear and self-contained result that supports downstream arguments without relying on unstated constraints.
>
> > Q5. In Theorem 4.2, Corollary 4.3 & Corollary 4.4 - why is the assumption on the eigenvalue distribution different per statement?
>
> **A**: The assumptions in Corollary 4.3 and Corollary 4.4 are specific instantiations of the more general condition stated in Theorem 4.2. We provided detailed derivations in Appendix A.3, showing how each corollary follows from the general theorem under simplified eigenvalue distributions.
>
> In particular, the convergence behaviour for the Mamba state is obtained by taking the limit $\lambda_{\min}⁡ \to 0$. For Mamba2, we consider the regime where $\lambda_{\max} \to \lambda_{\min}$​. These are chosen to reflect the parameterizations used in each model. We will clarify this by presenting the assumptions in formal mathematical notation and revise the manuscript accordingly.
>
> > Limitations: The empirical benefit of scaling instead of isn't clear. [...] It would be good to discuss [...] the tradeoffs and benefits of the different scaling methods.
>
> **A**: In future revisions, we will compare the tradeoffs of different scaling strategies and clarify their benefits, especially in comparison to MambaExtend. Corollaries 4.3/4.4 suggest that both eigenvalue adjustment and discretization step scaling can influence the convergence rate. While performance can vary depending on the model and sequence length, our results indicate that the scaling method becomes increasingly important as sequence length grows. We posit that scaling $A$ can be a dominant factor for extremely long sequences, where numerical stability and convergence speed become critical.
>
> > Paper Formatting Concerns: The paper has multiple flaws in the form of unclear sentences and other formatting issues, below are a few:
>
> **A**: Thank you for the detailed feedback, we will fix these typos in the final version.

---

> > ### Comment · Reviewer_z8Ky · 2025-08-09
> > **Response to rebuttal**
> >
> > Thank you for addressing my concerns.
> >
> > I have revised my position on the paper to borderline; however, given that other reviewers see merit in the work and consider it above the acceptance threshold, I will not argue for a contrary decision.

---

> > > ### Author Response · Authors · 2025-08-09
> > >
> > > We are elated that our response has addressed your concerns.
> > >
> > > We appreciate that you have raised your score; nevertheless, as you mention there are reasons for which you maintain a borderline position, we would be grateful if you have the time at any point to elaborate further on these such that we can provide additional and sufficient support either now or for a revised manuscript.
> > >
> > > Regardless, we again appreciate your engagement throughout this process and look forward to updating our work with your support.

---

### Official Review · Reviewer_ZsCU · 2025-06-20

**Clarity:** 4
**Significance:** 3
**Originality:** 3
**Rating:** 5
**Confidence:** 4

**Summary:**

In Mamba styled models (Mamba and Mamba2), there are 2 parameters which have so far been controlled for to enable length generalisation. One a state transition matrix - $A$, the other being $\Delta$, a parameter used for discretising the other main parameters in the SSM core equation. Prior work has focussed for controlling for $\Delta$, this paper argues why controlling $A$ would make much more sense. There are some theoretical results that the paper contributes that point to the same and the authors also provide quite strong experimental results to validate the claim.

**Questions:**

**Questions**
- Why does section 4.1 exist, what’s the point of that analysis ? This just lists the graph and says a generic statement, is there something I missed ? I currently think that not much will be lost if that section just never existed.
- Although 6.1 is very convincing, just maybe add a remark about why there could still be cases like GovReport Mamba - 1.4B where perplexity of scaling Delta is better. Acknowledge it or just share intuition.
- Based on the graphs, it seems like initially controlling for $\Delta$ starts out as being better, and then controlling for $A$ becomes better. Is there any scope of controlling for both, or is that stupid ?
- What do you think about the suggestion to rewrite the paper to be tighter / shorter to communicate the core idea quicker and more intuitively, or do you think this long paper format of NeurIPS makes most sense ?

**Suggestions**
- MambaExtend is by Azizi et al, that wasn’t clear from the main paper when it was mentioned, maybe add the citation in the section as well, instead of it just being in the Appendix.

**Ethical Concerns:**

["NO or VERY MINOR ethics concerns only"]

**Final Justification:**

I think the paper makes a minor albeit important point quite convincingly with quite extensive experimentation. My only concern before the rebuttal was regarding the length of the manuscript and the choice of a venue like NeurIPS. But the new experiments the authors have posted during the rebuttal as well as the discussions they have promised to include in the main paper would constitute a good paper in my view, and therefore I have increased my score to 5.

**Limitations:**

yes

**Quality:**

3

**Strengths And Weaknesses:**

**Strengths**
- There is very strong empirical evidence and the experiments are all very comprehensive.
- The core idea communicated in the paper is quite actionable and is thus quite relevant for LLM practitioners who use Mamba based foundation models.

**Weaknesses**
- The core idea communicated in the paper is quite simple, and I somewhere feel the writing of the paper currently has been optimised to fit the full 9 pages somehow. I think the paper could have been written in a much more succinct manner.

I am a bit conflicted about the paper. While I think the paper is quite solid and makes a very specific important point very convincingly, I do think that the full 9 pages are not required to put this point across. Thus, probably there is scope to think about a more appropriate format / venue for the paper. Although I wouldn't be opposed to this paper being accepted if others as well as the authors disagree. Especially because I don't think there is anything else that can make the paper better. Thus, I would be willing to raise my score further, in case the authors engage and communicate why they think the current format and version of the paper is a good choice to communicate this idea.

---

> ### Author Rebuttal · Authors · 2025-07-30
>
> We appreciate that the author acknowledges our work in terms of the evidence we provide for **our claims are convincing**, that **our paper is solid** and **the actionable nature of our ideas**. We also appreciate the various questions and suggestions they make, which we hope are resolved through the rebuttal that follows.
>
> >**C1**: I think the paper could have been written in a much more succinct manner. Do you think this long paper format of NeurIPS makes most sense?
>
> **A1**:  We understand the concern regarding the length and format, and we would like to take this opportunity to explain why we believe the current 9-page version is appropriate and necessary to fully communicate the significance of our contribution.
>
> **Shorter page limits may reduce persuasiveness.** There is a fine line between communicating an idea succinctly/effectively, while also providing sufficient justification through analysis and experiments. Thus, it is difficult to conclude regarding a shorter page limit, as we wish to maintain the same level of detail and persuasiveness.
>
> **A heavier appendix may reduce readability.** We already carefully offload proofs and more ancillary experimental and implementation details to appendix sections, allowing the main text to stay focused and readable. Condensing more main content could risk shifting essential arguments into the appendix, impacting overall readability and accessibility.
>
> **Additional analysis provided.** We appreciate your feedback and would like to add more analysis to enhance our work. We can provide the norm analysis of the `ssm_state` to enhance section 4.1 by conducting more experiments to record the trend of the `ssm_state` as the input sequence length increases as we replied to reviewer up1U. We randomly sampled data from ProofPile and PG19, and measured the maximum difference between the largest and smallest `ssm_state` norms across all layers of the Mamba2-1.3B model.  The results, presented in the following table, clearly demonstrate a divergent trend in the ssm_state norm as sequence length increases. This empirical observation is consistent with our theoretical prediction and further validates our analysis.
>
> We feel the depth of analysis offered by the current format aligns well with the standards and expectations of the NeurIPS long paper. At the same time, we are also happy to discuss and willing to make changes based on this interaction period.
>
> | proof-pile | 1k | 2k | 4k | 8k | 16k | 32k | 64k |
> |-|-|-|-|-|-|-|-|
> | State norm (max) | 131.4489 | 174.6176 | 892.7114 | 1330.5321 | 1141.5253 | 1130.6537 | 1157.5122 |
> | State norm (min) | 0.0046   | 0.0030   | 0.0007   | 0.0007    | 0.0004    | 0.0003    | 0.0005    |
> --------
> | PG19 | 1k | 2k | 4k | 8k | 16k | 32k | 64k |
> |-|-|-|-|-|-|-|-|
> | State norm (max) | 158.7656 | 160.8465 | 166.3788 | 876.0833 | 1241.3979 | 1201.5530 | 1242.7589 |
> | State norm (min) | 0.0033   | 0.0014   | 0.0050   | 0.0004   | 0.0005    | 0.0003    | 0.0002    |
> --------
>
> > **C2**: Why does section 4.1 exist, what’s the point of that analysis?
>
> **A2**: Section 4.1 motivates the theoretical developments in Section 4.2 by examining the numerical structure of the Mamba transition matrix.
>
> **Section 4.1 provides essential empirical motivation for the theory in Section 4.2.** The heatmap in Figure 1 reveals a consistent spectral pattern across layers: the coexistence of both large (near 1) and small (near 0) eigenvalues. This observation is crucial, as it sets the foundation for the theoretical analysis that follows—namely, the identification of two distinct dynamical regimes that govern the behavior of Mamba states.
>
> **Without this empirical grounding, the theory would lack connection to real model behavior.**  As mentioned above, we will add the state norm analysis to further ground the following theoretical section 4.2. This will make the connection between empirical trends and theoretical insights even more concrete in the final version of the paper.
>
> > **C3**: Although 6.1 is very convincing, just maybe add a remark about why there could still be cases like GovReport Mamba-1.4B where the perplexity of scaling Delta is better. Acknowledge it or just share intuition.
>
> **A3**: One way of interpreting this behavior is that at shorter lengths (ex. <= 16k), simply being able to scale either factor is sufficient, as the differences in perplexity are marginal. This may be because at these lengths, the compounded decay may approximate rather well; however, as the lengths keep increasing, this eventually leads to a significant difference between them, which may be the case of why at shorter lengths both methods appear to work rather well.
>
> On one note, we can observe that on Mamba (first generation) models, scaling $\Delta$ usually works slightly better on shorter lengths; we hypothesize that this might be due to how the $A$ matrix is constructed as a diagonal matrix rather than a scalar-times-identity like in Mamba2, which may make scaling $A$ with a single layer-specific scaling factor more appropriate in that scenario.
>
> One specific outlier is on Mamba-1.4b with GovReport, at a sequence length of 32k; scaling $\Delta$ for this example works significantly better than $A$ but it is the only case. We believe that this is more of a point of acknowledgement given the lack of other settings where this occurs (either with 1.4B models, first-generation Mamba models or just with other models on GovReport) and will mention this accordingly.
>
> > **C4**: It seems like initially controlling for $\Delta$ starts out as being better, and then controlling for $A$ becomes better. Is there any scope of controlling for both, or is that stupid?
>
> **A4**: There are manners in which this is possible; for example, if one is to introduce scaling parameters for each factor and train them in the same manner as we do, then by all accounts it is possible to control them simultaneously.
>
> **Characteristics of zeroth-order optimization**: One reason why this isn’t done explicitly is the use of zeroth-order optimization, which is known to become more unstable as the number of parameters that need to be tuned increases. Alternatively, one way would be to directly fine-tune everything through gradient descent, but this is a much more time- and resource-consuming process compared to the proposed design.
>
> **Multi-stage optimization**: Another method would be to conduct a type of multi-stage optimization, where perhaps one subset of scaling factors is tuned, followed by another. However, there are some known potential issues with such types of methods in deep learning research that make this somewhat difficult.
>
> > **C5**: MambaExtend is by Azizi et al, that wasn’t clear from the main paper when it was mentioned, maybe add the citation in the section as well, instead of it just being in the Appendix.
>
> **A5**: We thank the reviewer for raising this detail; we did mention Azizi et al. at multiple times within the preceding sections but did not explicitly mention MambaExtend; we will make this very clear by mentioning them together.

---

> > ### Comment · Reviewer_ZsCU · 2025-08-01
> >
> > I have read through the reviews of others and the responses of the authors to each of the reviews. I understand the choice of the format, and with the new experiments as well as some added discussion, I buy the argument that, the current format should be suitable.
> > The additional experiments on state norm are interesting and seem useful and would be a valuable addition to the current manuscript. Similarly the motivation of Section 4.1 and how that is useful for the rest of the paper is now clear to me, and I would recommend also including a bit more exposition of that in the main manuscript as well. It would also be good for the authors (as they have already agreed to) to provide a remark about experiments on Mamba-1.4B on GovReport, as it is an important outlier.
> >
> > If scaling both is possible in some cases and not possible in other cases, then it seems like a critical detail and remark that should be mentioned and discussed in detail, as it is quite a natural followup/ idea that one thinks of when reading the paper (reviewer vqA3 asked about it as well). I am not sure what amount of work is required to provide a proper answer to that question, the current argument seems a bit hand wavy, but I understand the intuition and also acknowledge that it might require a lot of effort to carefully investigate that. I would still recommend the authors to at least remark/ discuss this in the main paper.

---

> > > ### Author Response · Authors · 2025-08-04
> > >
> > > > It would also be good for the authors (as they have already agreed to) to provide a remark about experiments on Mamba-1.4B on GovReport, as it is an important outlier.
> > >
> > > **A**: We also will certainly include the specific details of the outliers; after the rebuttal, we have in fact found a hyperparameter setup that does appear to enable $A$ scaling to perform better/on-par with $\Delta_t$ scaling. To ensure a fair comparison for $\Delta_t$ scaling, we are currently verifying if we can in fact determine a better configuration for this setting as well. However, we promise to make note of the importance of these hyperparameters as a potential point of interest.
> > >
> > > |Length| $A$ Scaling | $\Delta_t$ Scaling |
> > > |-|-|-|
> > > | 16k | 5.1875 | 3.625 |
> > > | 32k | 6.03125 | 6.78125 |
> > >
> > > > I am not sure what amount of work is required to provide a proper answer to that question, the current argument seems a bit hand wavy, but I understand the intuition and also acknowledge that it might require a lot of effort to carefully investigate that. I would still recommend the authors to at least remark/ discuss this in the main paper.
> > >
> > > **A**: With respect to tuning $A$ and $\Delta_t$ simultaneously, we agree that there are merits to this, though there may be more thoughtful consideration necessary. It is true that strictly as an implementation detail, extending our current method to incorporate such a feature is rather straightforward. However, we have observed (under a number of additional setups) that in practice this does not directly work as well in practice. Currently, we have experimented on both perplexity evaluation and passkey retrieval. We have also tried a number of different ways of tuning each factor, for example using layer-wise scaling factors or per-state scaling factors for either $A$ or $\Delta_t$. Our observations here indicate that in neither case does this simultaneous tuning perform better than simply scaling one of these, with various instances of instability as well.
> > >
> > > We however do not view this as a negative result; rather, we believe that it is simply an indication that scaling both factors simultaneously may require more careful consideration, which is natural given the interactions between the two elements during computation. This thus remains as a follow-up, but it renders the problem more interesting as it may require considering additional theoretical/empirical factors that may have not yet been discovered as necessary in the current setting.

---

> > > > ### Comment · Reviewer_ZsCU · 2025-08-04
> > > >
> > > > I thank the authors for their response and I am satisfied with their response, and would like to increase my score.

---

> > > > > ### Author Response · Authors · 2025-08-05
> > > > >
> > > > > We are very grateful that our responses have resolved your outstanding questions and concerns. We remain very appreciative of your engagement and promptness throughout this process. If there remains any additional questions or suggestions, we are happy to provide details to answer these with the time that remains

---

### Official Review · Reviewer_up1U · 2025-07-02

**Clarity:** 3
**Significance:** 3
**Originality:** 3
**Rating:** 4
**Confidence:** 4

**Summary:**

This paper focuses on the sequence length extrapolation problem in Mamba-family models, identifies the spectrum of the transition matrix A as a key influencing factor, and proposes methods to adjust the spectrum of A to enhance the sequence extrapolation capability of Mamba-family models. The paper achieves promising experimental results.

**Questions:**

See the weaknesses mentioned above.

**Ethical Concerns:**

["NO or VERY MINOR ethics concerns only"]

**Final Justification:**

The authors have provided detailed explanations addressing the two specific concerns I raised. I believe they have clarified these issues to a certain extent. However, since my initial rating was already positive and not negatively affected by these matters in the first place, I’m inclined to maintain my current positive rating. Furthermore, what prevents me from further elevating my rating is that the state norm validation experiments—which I consider quite significant (as failure to validate might risk undermining the entire work's interpretation) —were only supplemented after being prompted by reviewers. This affects my fundamental assessment of the paper's thoroughness and rigor.

**Limitations:**

Yes.

**Paper Formatting Concerns:**

No format problem.

**Quality:**

3

**Strengths And Weaknesses:**

Strengths:

1. The problem addressed in this paper is important, and the proposed ideas are highly insightful.
2. The method proposed in this paper is easy to understand and implement.
3. The paper presents solid experimental results that offer valuable insights.

Weaknesses:

1. The paper argues that the instability (explosion or vanishing) of the state norm with increasing sequence length in Mamba-family models is the main cause of failure in length extrapolation. However, there is no direct experimental validation of the state norm behavior throughout the paper, which raises questions about the validity of the theoretical analysis.
2. Prior works, such as DeciMamba, have also proposed explanations for the failure of Mamba models in handling length extrapolation. This paper does not sufficiently argue that its explanation is superior to those provided in prior work, nor does it point out any flaws or inaccuracies in the existing explanations. This weakens the persuasiveness of the theoretical analysis.

---

> ### Author Rebuttal · Authors · 2025-07-30
>
> We thank the reviewer for the time and effort they have placed towards providing a thorough evaluation of our work, particularly their comments about **the problem being important**, **the experiment being solid**, and **both the ideas and results being insightful**. We are also appreciative of the weaknesses that have been raised and hope the following additional details/results will clarify these points and resolve these existing doubts about work.
>
> >**C1**: There is no direct experimental validation of the state norm behavior throughout the paper, which raises questions about the validity of the theoretical analysis.
>
> **A1**: We can fully address this concern by conducting additional experiments that track the behavior of the `ssm_state` as the input sequence length increases.
>
> Specifically, we randomly sampled data from ProofPile and PG19, and measured the maximum difference between the largest and smallest `ssm_state` norms across all layers of the Mamba2-1.3B model. The results, presented in the following table, clearly demonstrate a divergent trend in the `ssm_state`norm as sequence length increases. This empirical observation is consistent with our theoretical prediction and further validates our analysis.
>
> | ProofPile | 1k | 2k | 4k | 8k | 16k | 32k | 64k |
> |-|-|-|-|-|-|-|-|
> | State norm (max) | 131.4489 | 174.6176 | 892.7114 | 1330.5321 | 1141.5253 | 1130.6537 | 1157.5122 |
> | State norm (min) | 0.0046   | 0.0030   | 0.0007   | 0.0007    | 0.0004    | 0.0003    | 0.0005    |
> --------
>
> | PG19 | 1k | 2k | 4k | 8k | 16k | 32k | 64k |
> |-|-|-|-|-|-|-|-|
> | State norm (max) | 158.7656 | 160.8465 | 166.3788 | 876.0833 | 1241.3979 | 1201.5530 | 1242.7589 |
> | State norm (min) | 0.0033   | 0.0014   | 0.0050   | 0.0004   | 0.0005    | 0.0003    | 0.0002    |
> --------
>
>
> We appreciate your suggestion and will incorporate this analysis into the final version of the paper, as we agree it provides a valuable supplement that reinforces the strength of our contributions.
> > **C2**: Prior works, such as DeciMamba, have also proposed explanations for the failure of Mamba models in handling length extrapolation. This paper does not sufficiently argue that its explanation is superior to those provided in prior work, nor does it point out any flaws or inaccuracies in the existing explanations. This weakens the persuasiveness of the theoretical analysis.
>
> **A2**: Thanks for your feedback. While prior works, including DeciMamba, LongMamba, and MambaExtend, provided useful explanations for Mamba’s failure in long-sequence generalization, we believe our theoretical analysis offers a more comprehensive and principled understanding.
>
> Specifically, our contributions extend existing explanations in three key ways:
>
> - **New insight – divergent states due to large eigenvalues**: Our analysis reveals that large eigenvalues can lead to the tendency of exploding state norm, introducing instability that is not captured by existing explanations. This insight identifies an additional failure mode beyond vanishing memory and provides a more complete theoretical account of why Mamba models struggle with length extrapolation.
> - **Consistency with prior insights**: Our framework also recovers and formalizes the core idea shared by previous works: that small eigenvalues, when combined with accumulated discretization steps $\sum\Delta_{t}$, cause the decay term $\exp(-A \sum \Delta_t)$ to vanish, resulting in fast memory loss. In this sense, our theory subsumes and strengthens existing intuitions about the vanishing states.
> - **Unified and explanatory framework**: Through Corollaries 4.3 and 4.4, we show how modifying the discretization step $\Delta_{t}$ helps modulate convergence rates. However, we further clarify that such adjustments do not address the root cause for extremely long inputs, which lies in the spectral properties of the state transition matrix A. Our framework provides a principled explanation for why these heuristics sometimes succeed—and when they may fall short.
>
> In DeciMamba, the claims are quite complementary to our analysis. For example, they also make the claim about the cumulative product of the per-timestep transitions converging to 0; they do not provide any additional intuition as to why this may be the case. Our analysis, meanwhile, delves deeper into this phenomenon. While we don’t provide a subjective quantitative measure as they do, our inspection of the eigenvalue spectrum is meant as a way of highlighting that our claims have grounding and serves to motivate the rest of our analysis in Section 4. We hope this clarification addresses your concern and strengthens the persuasiveness of our theoretical contribution.

---

> > ### Comment · Reviewer_up1U · 2025-08-06
> >
> > Thank you for your clarifications, which have helped me better understand this work. The authors have specifically addressed the concerns I raised, and I believe they have provided reasonable clarification on these issues. Therefore, I maintain my current positive rating.

---

> > > ### Author Response · Authors · 2025-08-06
> > >
> > > Thank you for your engagement during this period. It delights us that our rebuttal/response has addressed your valid concerns that were raised within your review. Given that this response has, as you state, provided sufficient clarification for your existing questions, we would like to kindly ask if you could consider updating your score in a positive direction to reflect this. Should there be any further questions on your end, however, we would be more than happy to engage in additional discussions to resolve these in order for you to consider this.

---

> > > > ### Comment · Reviewer_up1U · 2025-08-06
> > > >
> > > > Thank you for your further response. We would like to clarify that during our initial evaluation, we did not automatically assume the mentioned issues necessarily existed and gave negative rating. Our original intention was to avoid prematurely assigning incorrect negative rating due to potential misunderstandings. Therefore, after careful discussion and clarification, we maintain our original positive rating.

---

> > > > > ### Author Response · Authors · 2025-08-06
> > > > >
> > > > > We appreciate the clarification and transparency; we respect the decision and are grateful for the positive sentiment towards our work. Nevertheless, if any further questions or comments arise during the time that remains, we are more than willing to provide additional details as needed.

---

### Official Review · Reviewer_vqA3 · 2025-07-03

**Clarity:** 2
**Significance:** 2
**Originality:** 1
**Rating:** 4
**Confidence:** 4

**Summary:**

This paper identifies spectral instability in the state-space transition matrix A as the cause of weak length generalization of Mamba & Mamba-2.
The authors prove that extreme eigenvalues critically affect hidden-state divergence. To remedy this problem, this paper proposes uniformly rescaling A with calibrated values through backpropagation.
This scaling of A is training-free (using pretrained weights directly) without updating model weights (but it actually tunes A values through calibration).
This solution keeps reasonable perplexity upto 64 K–128 K tokens while boosting long-context accuracy on tasks such as LongBench and Passkey Retrieval.

**Questions:**

Does this calibration method actually perform better than fine-tuning only A or $\Delta$t values (with other parameters fixed) directly to long context length?

What happens if combining both $\Delta$t and A scaling?

**Ethical Concerns:**

["NO or VERY MINOR ethics concerns only"]

**Final Justification:**

I appreciate the authors' detailed response. After reading the reviews and the responses, I believe the experimental results added during the rebuttal make the paper more promising, thus I raise my score. As the authors mention in their response, the experimental results (e.g., scaling both dt and A) do not necessarily mean negative results, and they rather help readers understand the value of the work. So, I strongly recommend the authors to add discussions and results in the main paper.

**Limitations:**

yes

**Paper Formatting Concerns:**

no concerns

**Quality:**

2

**Strengths And Weaknesses:**

The idea is simple, and the calibration requires very minimal training of multiplier values, which leads to good reproducibility. The results are competitive, maintaining good perplexity.
However there are some weaknesses:
- the contributions are limited : scaling a different parameter in the same training-free calibration framework has only subtle differences from MambaExtend.
- the theoretical analysis assumes uniform, diagonal spectra, limiting real-world applicability.
- it is not clear how section 4.2 directly leads to the motivation of controlling A only.
- using one scalar per layer coarsely addresses spectral outliers. this leaves finer eigenvalue issues not fully resolved


minor: figure 2 legend color should be corrected.

---

> ### Author Rebuttal · Authors · 2025-07-31
>
> Thank you for your valuable feedback. We greatly appreciate your recognition of **our framework's efficiency**, **good reproducibility** and **competitive results**. We nevertheless appreciate the additional points that have been mentioned as limiting factors and hope that the following response will to address these follows:
>
> ---
>
> >C0: This scaling of A is training-free (using pre-trained weights directly) without updating model weights (but it actually tunes A values through calibration).
>
> **A0**: To clarify, our method introduces a lightweight set of scaling factors applied to the pre-trained model; the number of scaling factors is smaller than the number of A. While it is possible to tune the entire A, our method and experiments avoid doing so. The scaling set functions as a `plug-and-play` module that can be activated only when the input sequence length exceeds the model's original maximum context length during inference.
>
> ---
>
> >C1: The contributions are limited : scaling a different parameter in the same training-free calibration framework has only subtle differences from MambaExtend.
>
> **A1**: We would like to highlight our contribution and distinguish our work from previous studies.
>
> **Novel Theoretical Analysis.** We are the first to provide a theoretical explanation for Mamba’s failure to generalize to long sequences, identifying the root cause as the divergent growth of the state norm due to spectral properties of the state transition matrix. This insight is not only novel but also foundational, as it enables a deeper understanding of the underlying dynamics rather than relying on heuristic modifications.
>
> **Strong Interpretability.** While we recognize the simplicity and practicality of the scaling methods in MambaExtend, our approach is theoretically motivated and analytically grounded. In particular, our analysis explains why certain empirical scaling factors (e.g., those greater than 1) observed in MambaExtend emerge despite appearing counterintuitive under their original motivation.
>
> ---
>
> >C2: The theoretical analysis assumes uniform, diagonal spectra, limiting real-world applicability.
>
> **A2**: We could fully clarify your concerns regarding the uniform and diagonal assumption.
>
> **1. The premise that our theoretical analysis assumes diagonal spectra is a point of misunderstanding.** In fact, the diagonality is not a simplifying assumption—it is a structural characteristic of Mamba and Mamba-2. Their state transition matrices are inherently diagonal, which means the spectrum and the matrices of the form $\text{diag}(\exp(-A))$ naturally share identical entries. We don’t impose any additional assumptions on them.
>
> **2. Our choice to model the spectrum using a uniform distribution is empirically grounded.** As shown in Section 4.1, the eigenvalues of pretrained Mamba models consistently span the range from 0 to 1 across all layers. This observed behavior supports our decision to use a uniform distribution as a practical and realistic approximation.
>
> **3. The divergent convergence behavior is irrespective of spectral assumptions.** We appreciate your rigorous comment and accordingly extend our analysis to accommodate a more generalized case without imposing any strict assumptions on the distribution of $\lambda$. Notably, the asymptotic behavior of the system remains unchanged.
>
> ---
>
> **Elaboration on Theorem 4.2**: Expected State Norm under General Spectral Distribution
>
> Assume the transition matrix $\mathbf{\Lambda} = \mathrm{diag}(\lambda_1, \dots, \lambda_d)$ has diagonal entries $\lambda_i \in (0,1)$ drawn i.i.d. from a probability density function $p(\lambda)$ supported on $[\lambda_{\min}, \lambda_{\max}] \subset (0,1)$.
>
> Suppose the system evolves as:
> $\mathbf{h_t} = \mathbf{\Lambda} \mathbf{h}_{t-1} + \mathbf{B} \mathbf{x}_t,$
> where $\mathbf{x}_t \sim \mathcal{N}(0, I)$ and each row of $\mathbf{B}$ is independently sampled as $\mathbf{b} \sim \mathcal{N}\left(0, \frac{1}{\sqrt{d}} \mathbf{I}\right)$.
>
> Then, as $t \to \infty$, the expected squared norm of the hidden state converges to
> $
> \boxed{
> \mathbb{E}[\|\mathbf{h_\infty}\|^2] = \mathbb{E}[\|\mathbf{B} \mathbf{x}\|^2] \cdot \int_{\lambda_{\min}}^{\lambda_{\max}} \frac{p(\lambda)}{1 - \lambda^2} \, d\lambda.
> }
> $
>
> ---
>
> **Asymptotic Analysis of $\mathbb{E}[\|\mathbf{h}_\infty\|^2]$**
>
> Define $
> I(\lambda_{\min}, \lambda_{\max}) := \int_{\lambda_{\min}}^{\lambda_{\max}} \frac{p(\lambda)}{1 - \lambda^2} \, d\lambda.
> $
>
> We analyze the behavior of $I(\lambda_{\min}, \lambda_{\max})$ as $\lambda_{\max}$ approaches the edges of the domain.
>
> 1. Behavior as $\lambda_{\max} \to 1^{-}$:
>
>  Near $\lambda = 1$: , we have $ 1 - \lambda^2 \approx 2 (1 - \lambda)$ via Taylor expansion.
>
>  If $p(\lambda)$ is continuous at $\lambda=1$ and $p(1) > 0$, then for $\lambda_{\max}$ close to 1, $I(\lambda_{\min}, \lambda_{\max}) \approx \frac{p(1)}{2} \int_{\lambda_{\min}}^{\lambda_{\max}} \frac{1}{1 - \lambda} \, d\lambda = \frac{p(1)}{2} \log\left(\frac{1 - \lambda_{\min}}{1 - \lambda_{\max}}\right).$
>
>  As $\lambda_{\max} \to 1^{-}$, $\log(1 - \lambda_{\max}) \to -\infty$, so
> $\boxed{I(\lambda_{\min}, \lambda_{\max}) \sim -\frac{p(1)}{2} \log(1 - \lambda_{\max}) \to +\infty.}$
>
> The expected state norm diverges logarithmically as the largest eigenvalue approaches 1.
>
>  2. Behavior as $\lambda_{\max} \to 0^{+}$
>
>  When $\lambda_{\max}$ is close to 0,
> $
> \frac{1}{1 - \lambda^2} \approx 1,
> $
> since $\lambda^2$ is very small.
>
>  If $p(\lambda)$ is continuous near zero with $p(0) > 0$, then
> $I(\lambda_{\min}, \lambda_{\max}) \approx \int_{\lambda_{\min}}^{\lambda_{\max}} p(\lambda) \, d\lambda \approx p(0) \cdot \lambda_{\max}.$ Hence, $\boxed{I(\lambda_{\min}, \lambda_{\max}) \to 0 \quad \text{linearly as } \lambda_{\max} \to 0^{+}}$.
>
> The expected state norm vanishes linearly with $\lambda_{\max}$ approaching zero.
> `The expanded analysis still reveals the inherent tendencies of Mamba’s state dynamics toward explosion or vanishing, irrespective of spectral assumptions.`
>
> ---
>
> > C3: It is not clear how section 4.2 directly leads to the motivation of controlling A only.
>
> **A3**: The eigenvalues of the transition matrix is $diag(\exp(-A))$ , thus we can control the spectrum distribution by applying a scaling factor to control the magnitude of A accordingly.
>
> > C4: Using one scalar per layer coarsely addresses spectral outliers.
>
> **A4**: For fair comparison, we follow the default setting of MambaExtend for both language modeling and NIAH experiments. The experiments on the language modeling task are applied with per-layer-wise scaling. We could perform state-wise scaling accordingly; however, this will lead to more computation compared to MambaExtend methods, as it will introduce more trainable parameters.
>
> On one note, the NIAH results do use state-wise scaling for each layer for MambaExtend and ours, with the end results similar to what we observe on other datasets. We contend that this serves as additional evidence to support the proposed method of scaling $A$ as opposed to $\Delta$ is valid.
>
> > C5: Does this calibration method actually perform better than fine-tuning only A or t values (with other parameters fixed) directly to long context length? What happens if combining both t and A scaling?
>
> **A5**: We note that the goal of using zeroth order optimization method here is intended to make calibrating the model significantly simpler and faster, as it only requires forward passes and no explicit gradient. Scaling both $A$ and $\Delta_t$ simultaneously can possibly lead to better results, but at the cost of greater complexity and potential instability as zeroth-order optimization often relies on tuning a small set of parameters.
>
> Furthermore, $\Delta_t$ is input-dependent, not a model parameter, and therefore cannot be directly fine-tuned.
> We manage to add fine-tuning $A$ directly and calibrate full $A$ for language modeling on PG19 with Mamba models. Enable more tuning freedom with additional scaling factors can improve the performance accordingly.
> | PPL | 2k | 4k | 8k | 16k | 32k | 64k |
> |-|-|-|-|-|-|-|
> | **mamba-1.4b** | | | | | | |
> | Base | 9.67   | 10.23  | 11.43  | 17.46  | 59.77   | 444.09   |
> | MambaExtend         | **4.69** | **3.89**  | **3.83**  | **3.55**  | 5.31   | 1648.00   |
> | Calibrated $A$ | 5.31 | 4.31 | 4.13 | 6.88 | 14.94 | 19.13 |
> | Finetuning $A$ (full)| 5.03 | 5.09 | 4.72 | 4.13 | 5.69 |  **6.45** |
> | Calibrated $A$ (full)| 5.28 | 4.38 | 4.42 | 4.63 | **5.16** |   6.48 |
> | **mamba2-1.3b** | | | | | | |
> | Base | 9.52   | 10.54  | 25.49  | 115.65 | 634.32  | 1479.45  |
> | MambaExtend | **4.34**  | 3.69  | 3.44  | 5.10   | 14.94   | 27.50 |
> | Calibrated $A$ | 4.38 | 3.78 | 3.44 | **3.28** | 4.03 | 4.72 |
> | Finetuning $A$ (full)  | 4.41 | 3.72 | **3.43** | 3.29 |  **3.23** |  **3.20** |
> | Calibrated $A$ (full)  | 4.38 | **3.68** | 3.52 | 3.44 |  **3.23** |  3.24 |
>
>
> The following is a full fine-tuning of only $A$ for the NIAH task, which we observe does not significantly deviate from directly scaling $A$ but remains nearly exactly comparable to our method of simply adding scaling parameters. Here, `X` indicates the model correctly solved all examples.
>
> **Mamba-1.4B**
>
> |Depth/Length| 1K | 2K | 4K | 8K | 16K | 32K |
> |-|-|-|-|-|-|-|
> |0%| X | X | X | X | X | X |
> |25%| X | X | X | X | X | X |
> |50%| X | X | X | X | X | - |
> |75%| X | X | X | X | X | X |
> |100%| X | X | X | X | X | X |
>
> We use the same evaluation criteria as described within the manuscript for all additional experiments.
>
> > Q: Figure 2 legend color should be corrected.
>
> **A**: Thank you for this note; the legend was intended to show that the changing $A$ scales were represented by different colors while the scaling of $\Delta_t$ used different line styles. We can understand where the confusion came about and will fix this issue.

---

### Note · Authors · 2025-08-11

We thank the reviewers for their engagement throughout the discussion process. We are pleased that each of them mentions that we have resolved their concerns (up1U, z8Ky, ZsCU) or that our additional results within the rebuttal have made the results more promising (vqA3).

To summarize our rebuttal, we have provided the following details:
- Further clarification of our contribution, which leverages new insight on the divergent states due to large eigenvalues to motivate a unified and explanatory framework that underlies our methodology; such analysis is novel and provides a more grounded approach to understanding length extrapolation in Mamba models.
- Additional clarification of Section 4.2, which builds a connection between the state norm and the eigenvalues of $A$, motivating our decision to scale $A$. We further provide explicit empirical results on the state norm to show that this becomes a concern in practice, validating our analysis.
- We provide additional results on the tuning of both $A$ and $\Delta_t$ simultaneously as well as additional per-state tuning results, showing that our results hold. Furthermore, results using a direct fine-tuning of either $A$ is provided as a baseline to show the superiority of the proposed method.
- Clarified additional details regarding relationships to prior works, such as clarifying prior existing works and prior motivations for existing methods.

Overall, this discussion period has yielded a productive and positive result. **However, there remains a borderline assessment expressed by reviewer z8Ky, who responded to our rebuttal just before the discussion deadline by admitting that our response addressed their concerns;** yet their statement suggests that they remain negative (`“given that other reviewers see merit in the work and consider it above the acceptance threshold, [they] will not argue for a contrary decision”`) with high confidence (4). However, there are ``no additional details`` regarding the reasoning behind this opinion and therefore we don't have an opportunity to provide a response to counter it.

We would again like to note that our responses have allowed multiple reviewers to raise their rating of our work and there are no additional concerns that the reviewers mention; accordingly, we hope this is sufficient to convince both the reviewers as well as the ACs of the merits of our work and therefore allow them to express a positive assessment of our work and update their ratings to reflect this.

---

### Decision · Program_Chairs · 2025-09-17

**Decision:**

Accept (poster)

**Comment:**

This paper proposes a method for improving the length generalization of Mamba models by scaling the spectrum of transition matrices. Reviewers found the approach simple, well-motivated, and supported by strong experiments and theoretical analysis. While some viewed the contribution as incremental relative to existing calibration methods, the rebuttal provided additional clarifications and results that addressed most concerns. Overall, the contribution is clear and well-supported, and the consensus is that the paper is suitable for acceptance.